# Anomalous circular bulk photovoltaic effect in BiFeO₃ thin films with stripe-domain pattern

David S. Knoche [1,2], Matthias Steimecke [3], Yeseul Yun[1,2], Lutz Mühlenbein[1,2] & Akash Bhatnagar [1,2✉]

Multiferroic bismuth ferrite, BiFeO₃, offers a vast landscape to study the interplay between different ferrroic orders. Another aspect which is equally exciting, and yet underutilized, is the possibility of large-scale ordering of domains. Along with symmetry-driven bulk photovoltaic effect, BiFeO₃ presents opportunities to conceptualize novel light-based devices. In this work, we investigate the evolution of the bulk photovoltaic effect in BiFeO₃ thin films with stripe-domain pattern as the polarization of light is modulated from linear to elliptical to circular. The open-circuit voltages under circularly polarized light exceed ± 25 V. The anomalous character of the effect arises from the contradiction with the analytical assessment involving tensorial analysis. The assessment highlights the need for a domain-specific interaction of light which is further analyzed with spatially-resolved Raman measurements. Appropriate positioning of electrodes allows observation of a switch-like photovoltaic effect, i.e., ON and OFF state, by changing the helicity of circularly polarized light.

[1] Zentrum für Innovationskompetenz SiLi-nano, Martin-Luther-Universität Halle-Wittenberg, Halle (Saale) 06120, Germany. [2] Institut für Physik, Martin-Luther-Universität Halle-Wittenberg, Halle (Saale) 06120, Germany. [3] Institut für Chemie, Technische Chemie, Martin-Luther-Universität Halle-Wittenberg, Halle (Saale) 06120, Germany. ✉email: akash.bhatnagar@physik.uni-halle.de

The re-discovery photovoltaic effect in ferroelectrics has far reaching implications that have been till now demonstrated. The obvious ones include the realization of junction free-photovoltaic modules[1], and tracing polarization components, both in-plane[2] and out-of-plane[3], via non-destructive methodologies. Generation of above bandgap open-circuit voltage ($V_{oc}$) under suitable illumination is another outcome which has garnered immense attention[1,4]. The underlying symmetry-driven charge-separation mechanism is even more enticing which has fueled research for materials with broken inversion symmetry. Conversely, the effect itself has been also utilized to detect the status of inversion symmetry in a wide range of materials such as oxides[5], topological insulators[6], and inorganic-organic perovskites[7,8]. In this context, the circular bulk photovoltaic effect (CBPV) is of particular interest. In general, the photovoltaic effect in materials without inversion symmetry can be resolved into two parts, linear and circular[5]. Much of the work till now has been focused on the linear bulk photovoltaic effect (LBPV) wherein the resulting photovoltaic current depends on the orientation of the linearly polarized (LP) light[5].

The CBPV, on the other hand, results in a photocurrent which depends on the helicity of the CP light[9–12]. An essential prerequisite for the manifestation of such an intriguing phenomenon, apart from the absence of inversion symmetry, is a split in the conduction or valence band of the material in reciprocal space, which is also known as Rashba and Dresselhaus effect[9,11,13]. As a result, light of a given helicity excites charges with spin up ↑, while of opposite helicity charges of spin down ↓. Consequently, the CBPV has been often utilized as probing methodology to detect Rashba splitting in different bulk material systems such as BiTeBr[14] and ZnO[15], and even in heterostructures comprising of AlGaN-GaN[16]. In addition, measurements of the CBPV have been instrumental in confirming the absence of inversion symmetry in different organic-inorganic perovskite-structured systems, which has assisted in the pursuit of finding the origin of charge separation mechanisms in these materials system[17].

Therefore, it is rather intriguing that the CBPV has never been investigated in BiFeO$_3$, although all the essential criteria are evidently satisfied. The rhombohedral polar space group R3c of bulk BiFeO$_3$ leads to ferroelectricity[18] and the existence of the LBPV. The crystal symmetry also allows gyrotropy and thus the CBPV should be observable in BiFeO$_3$[5]. This becomes more interesting with BiFeO$_3$ having a bandgap ($E_g \approx 2.7\,eV$)[19] which lies within the visible range of solar spectrum and a corresponding photovoltaic effect that has been shown to be governed by the symmetry.

The ferroelectric domain arrangement can be engineered in a periodic array by tuning growth conditions, such as substrate symmetry[20] and termination[21], gas pressure, and related composition[22]. The room temperature multiferroic character of BiFeO$_3$ manifests in a rather intricate coupling between the polarization vector and magnetic order[23]. Consequently, the control of the magnetic order by the application of electric fields has been predicted and demonstrated successfully in thin film-based devices. Furthermore, the magnetic order in BiFeO$_3$ is currently under intense investigation which exhibits a non-collinear antiferromagnetic spin cycloidal structure and is a potential candidate for future antiferromagnetic spintronic devices[24]. An in-depth view of the magnetic order was recently provided that explicitly illustrated the existence of spin cycloids within each domain having a propagation vector aligned orthogonal to the polarization[25]. Consequently, the periodic ferroelectric domain pattern essentially renders an equally ordered magnetic texture of BiFeO$_3$ thin films grown on a variety of different substrates[26]. This evidently presents some interesting opportunities to analyze domain-specific light–matter interactions, inherent to materials which have chiral textures[27]. Therefore, it becomes apparent that BiFeO$_3$ thin films provide a lucrative landscape to study the overlap of these rather discrete aspects.

In this work, we investigate the CBPV in epitaxially grown BiFeO$_3$ thin films. The LBPV is first utilized to establish the photovoltaic activity of the samples with $V_{oc}$ well above the bandgap. The BPV is investigated systematically by modulating the polarization of the light from linear to circular. A changed CP light helicity (left ↔ right) generates $V_{oc}$ of opposite polarities, albeit with nearly half of the magnitude in comparison to the maximum response under LP light. Analysis of the response with the CBPV tensor suggests a rather compelling scenario involving domain-specific interaction of the CP light. This aspect is tested with different measurement geometries which results in a switch-like state exhibiting a chirality-dependent ON and OFF state of the photovoltaic effect. Spatially-resolved Raman measurements, and related analysis, further bolster the arguments.

## Results

**Thin-film growth and characterization.** BiFeO$_3$ thin films with a thickness of around 200 nm were deposited on single-crystalline DyScO$_3$ (110)$_{orth}$ substrates using a pulsed laser deposition system. Further details on sample growth can be found in the Methods sections. The topography of the resultant samples appears to be smooth (root-mean-square roughness $R_q \approx 850\,pm$) with stripes along [100]$_{pc}$ as shown in Fig. 1a. The lateral signal, phase (Fig. 1b), and amplitude, acquired from piezo force

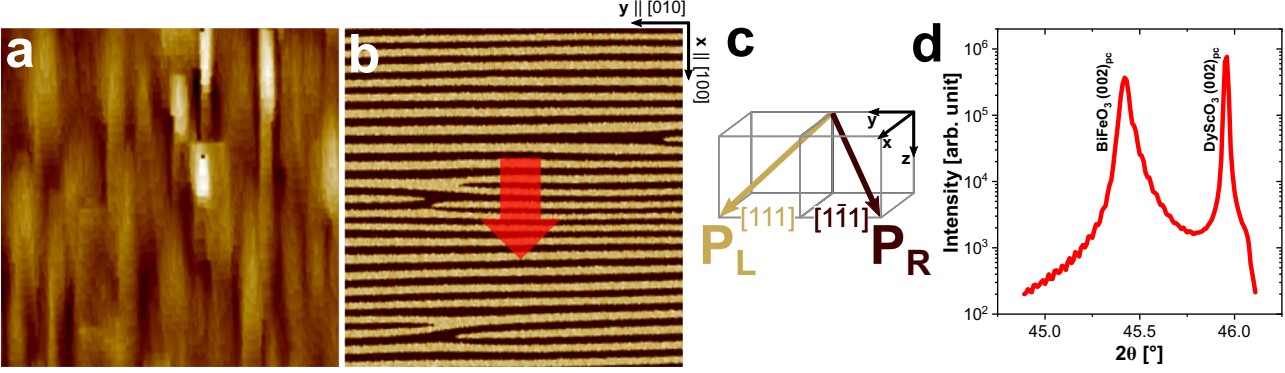

**Fig. 1 Domain and structural characterization of BiFeO$_3$ thin film. a** Surface morphology visualized using atomic force microscopy ($R_q \approx 850\,pm$) and **b** corresponding in-plane PFM phase image (cantilever aligned along [100]$_{pc}$, 7.5 × 7.5 μm²). Red arrow indicates the in-plane net polarization resulting from the stripe-ordered domain arrangement of only two polarization variant P$_L$ and P$_R$ shown schematically in **c**. **d** X-Ray diffraction $2\theta - \omega$ scan around (002)$_{pc}$ substrate peak.

microscopy (PFM) reveal a periodic domain pattern and the domains stretch across the entire width of the scan (7.5 µm) on many instances. The striped domain arrangement reveals an average period length around 170 nm in $[100]_{pc}$. As the vertical signal from PFM was largely consistent in color (Supplementary Fig. 1), the existence of only two variants, out of the eight possible, which are separated by 71° domain walls can be implied. This periodic arrangement of stripe domains lead to a resulting net polarization in $[101]_{pc}$. The crystallinity and phase purity of the samples were confirmed by X-ray analysis and a $2\theta - \omega$ scan around the $[002]_{pc}$ substrate peak is presented in Fig. 1d. Because of the mismatch between the lattice parameter of the film $a_{BFO,pc} = 3.965$ Å[18] and the substrate $a_{DSO,pc} = 3.946$ Å, $b_{DSO,pc} = 3.952$ Å[28] an anisotropic compressive strain is implied during the epitaxial thin film growth. The respective mismatch between the in-plane lattice constant of $-0.33\%$ and $-0.48\%$ leads to a resultant increased out-of-plane lattice constant of $BiFeO_3$ $c_{BFO,pc} = 3.991$ Å. The value is in agreement with other studies wherein, similar $c$ and strain were observed due to the largely identical in-plane lattice constants of the $BiFeO_3$ film and $DyScO_3$ substrates [20,26].

**Bulk photovoltaic effect.** The samples were first measured with LP light with a wavelength of 405 nm (3.06 eV). In Fig. 2a, the measurement geometry is schematically depicted with electrodes running along x-direction perpendicular to the domain walls. The generated photovoltaic current (density) $j_i^L$ for an ordered striped domain arrangement with 71° domain walls can be described with following equation (Details: Supplementary Fig. 2/Eqs. (1)–(15)):

$$j_i^L = I \begin{pmatrix} \left(\frac{\beta_{33}^L}{3\sqrt{3}} + \frac{2\beta_{31}^L}{3\sqrt{3}} + \frac{\beta_{22}^L}{3\sqrt{6}} + \frac{\beta_{15}^L}{6\sqrt{3}}\right) + \left(\frac{\beta_{22}^L}{\sqrt{6}} + \frac{\beta_{15}^L}{2\sqrt{3}}\right)\cos(2\theta) \\ \left(\frac{\beta_{33}^L}{3\sqrt{3}} - \frac{\beta_{31}^L}{3\sqrt{3}} - \frac{2\beta_{22}^L}{3\sqrt{6}} + \frac{\beta_{15}^L}{6\sqrt{3}}\right)\sin(2\theta) \\ \left(\frac{2\beta_{22}^L}{3\sqrt{6}} + \frac{\beta_{15}^L}{3\sqrt{3}} - \frac{\beta_{33}^L}{3\sqrt{3}} - \frac{2\beta_{31}^L}{3\sqrt{3}}\right) \end{pmatrix} \quad (1)$$

wherein $I$ is the light intensity, $\beta_{ij}^L$ are the LBPV coefficients, and $\theta$ is the angle describing the rotation of the electric field plane of the LP light around the z-axis. The rotation is achieved by using a half-wave $(\frac{\lambda}{2})$ plate.

The measured values of the photovoltaic current (density) in y-direction are largely similar to the previously reported values[1,2]. The dependency on the light orientation $\theta$ match qualitatively the predicted sinusoidal response in Eq. (1) and confirms the dominance of the BPV. In addition, the photovoltaic effect was found to be switchable in its characteristics as electric fields above the coercive field were applied across the electrodes. The resultant response largely mimics ferroelectric switching which further validates the dominance of ferroelectric/bulk photovoltaic effect in these samples (Supplementary Fig. 3). It must be emphasized here that the magnitude of photovoltaic current scales up linearly with the intensity of light. On the contrary, the $V_{oc}$ should remain constant above a threshold light intensity[29]. We also tested this condition by gradually scaling up the light intensity and observed a distinct saturation of $V_{oc}$ above a certain intensity of light. Furthermore, the current-voltage characteristics are linear with an unchanged slope for different light orientation $\theta$ (Supplementary Fig. 4/5) Therefore, both $j_i$ and $V_{oc}$ can be used to describe the observed photovoltaic effect, however, because of the insensitivity towards slight light intensity changes, from hereon, only the extracted values of $V_{oc}$ will be presented.

The photovoltaic current (density) originating from the CBPV $j_i^C$ depends on the helicity of the CP light (Details: Supplementary Eqs. (16)–(23)):

$$LCP : j_i^{LCP} = \frac{I}{\sqrt{3}} \begin{pmatrix} 0 \\ +\beta_{12}^C \\ 0 \end{pmatrix} \quad (2a)$$

$$RCP : j_i^{RCP} = \frac{I}{\sqrt{3}} \begin{pmatrix} 0 \\ -\beta_{12}^C \\ 0 \end{pmatrix} \quad (2b)$$

wherein $j_i^{LCP}$ and $j_i^{RCP}$ are the generated CBPV current density under illumination with left-handed circularly polarized (LCP) and right-handed circularly polarized (RCP) light, respectively. For both light chiralities, the current depends on the light intensity $I$ and the CBPV coefficient $\beta_{12}^C$. The current in x- and z-direction is zero, whereas in y-direction reverses its direction from LCP to RCP light.

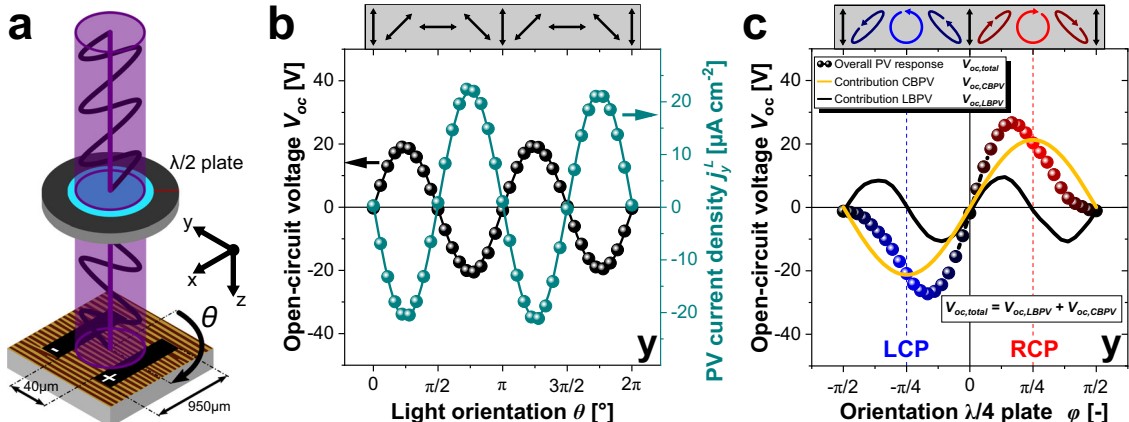

**Fig. 2 Photoelectrical measurements revealing LBPV and CBPV in BiFeO₃ thin films. a** Schematic of the experimental setup (not to scale): A $\frac{\lambda}{2}$ plate is used to rotate the orientation of the incoming LP light (rotation described by $\theta$). The light propagates in z-direction and the electrodes run along the x-direction to enable photoelectrical measurements in y-direction. **b** LBPV measurement: $V_{oc}$ and $j_y^L$ as a function of the light orientation $\theta$. The two-sided arrows indicate the orientation of LP light. **c** CBPV measurement: $\frac{\lambda}{4}$ plate placed before $\frac{\lambda}{2}$ plate to enable modification of the light polarization state. The flip of the light helicity due to the presence of the $\frac{\lambda}{2}$ plate is considered in the experimental data. $V_{oc}$ as a function of the $\frac{\lambda}{4}$ plate orientation $\phi$. The light polarization varies from linear ($\phi \in \{\pm\frac{\pi}{2}, 0\}$), to elliptical ($\phi \notin \{\pm\frac{\pi}{2}, \pm\frac{\pi}{4}, 0\}$) and to circular ($\phi \in \{\pm\frac{\pi}{4}\}$). The helicity of the light polarization changes from left-handed ($-\frac{\pi}{2} < \phi < 0$) (blue) to right-handed ($0 < \phi < \frac{\pi}{2}$) (red). The overall $V_{oc}$ can be divided into the contribution from the CBPV (yellow solid line) and the LBPV (black solid line).

A quarter-wave ($\frac{\lambda}{4}$) plate was used to change the light polarization. The angle between the the fast axis of the $\frac{\lambda}{4}$ plate and the polarization plane of incident LP light is defined as $\phi$. A variation in $\phi$ apparently modulates the circularity of the light following $P_c = \sin(2\phi)$. As $\phi$ is varied, the light polarization gradually changes from linear to elliptical, and eventually to circular at $\phi = \pm\frac{\pi}{4}$. The $V_{oc}$ as a function of $\phi$ is presented in Fig. 2c. The corresponding light polarization is shown in the schematic above. The $V_{oc}$ peaks at $\phi = \pm\frac{\pi}{6}$ with a corresponding magnitude of $\pm 27$V, which is in the regime of elliptical light polarization. The elliptical polarization can be considered as a sum of linear and circular polarization, and consequently, the net measured response is a sum of contributions from LBPV and CBPV. The CBPV response scales up linearly with $P_c$, which makes it possible to extract the respective contribution (yellow solid line, $V_{oc,CBPV}(\phi) = 21.3$V $\sin(2\phi)$) from the overall response. The LBPV contribution (black solid line) can be obtained by subtracting the CBPV contribution from the overall response. This was confirmed by an additional measurement conducted with the outcoming light polarization rotated by 90°. The extracted LBPV contribution is shifted while the CBPV contribution remains unchanged (Supplementary Fig. 6).

Similar responses have been also reported in Tellurium[9,11], in BiTeBr[14], and more recently in organic-inorganic halide perovskites[8]. In all of these materials systems, the extreme values appear at $\phi = \pm\frac{\pi}{4}$ corresponding to LCP and RCP light. This can be attributed to spin splitting and the excitation of carriers either in $k_x > 0$ or $k_x < 0$ by LCP and RCP light, respectively. The eventual relaxation of the carrier to the corresponding bottom and top of conduction and valence band, respectively, results in photocurrent of opposite direction under illumination with RCP and LCP light, respectively[8,30]. However, this not the case in our measurements wherein the maximum response appears at $\phi = \pm\frac{\pi}{6}$ because of the superimposed contribution from the LBPV.

**Linear to circular bulk photovoltaic effect.** In order to distinctly differentiate between the responses arising from LBPV and CBPV, we modified the setup by replacing the $\frac{\lambda}{4}$ plate with a Berek compensator (tunable wave plate). This allowed us to obtain light polarization with different ellipticity ratios $\varepsilon$ ranging from $\varepsilon \approx 1$ (CP light) to $\varepsilon > 300$ (LP light) without rotating the main axis of the light polarization ellipse ($\|[001]_{pc}$, see Fig. 3d). Details about the light polarization characterization can be found in Supplementary Fig. 7. A subsequent $\frac{\lambda}{2}$ plate enabled the clockwise rotation of the out-coming light by an angle $\theta$ (compare Fig. 2b).

In the first instance, identical geometry as depicted in Fig. 2 was used (electrodes $\|$ x-direction). Fig. 3b shows the $V_{oc}$ as a function of $\theta$ for light with different $\varepsilon$. The LP light ($\varepsilon > 300$) generates an expected sinusoidal response with maximum $V_{oc} = \pm 19.2$V (Compare Fig. 2b, Eq. (1)). Interestingly, the right-handed elliptically polarized (EP) light (with $\varepsilon = 3$, $\varepsilon = 10$) results in higher maxima of around 27.6V (Compare Fig. 2c), and the response is visibly less dependent on $\theta$. As $\varepsilon$ is further reduced to 1, the angle $\theta$ is rendered redundant and a consistent $V_{oc}$ of 20.6 V is observed. Conversely, this also signifies the precession of the CP light. As the helicity of light is changed from right to left, the response of generated $V_{oc}$ follows a similar trend, but with negative polarity. The results with this setup reveal a more detailed insight into the individual contributions arising from LBPV and CBPV. The results are also in coherence with those shown in Fig. 2c and Eq. (2a). The direction of the $V_{oc}$ arising from the CBPV depends on the helicity of the light polarization.

The same measurements were repeated with a different pair of electrodes which were fabricated to measure the response in x-direction. The LBPV response follows now a cosinusoidal relation, instead of sinusoidal, which agrees with Eq. (1). It exhibits higher $V_{oc}$ maxima of around 54V at $\theta \in \{\frac{\pi}{2}, \frac{3\pi}{2}\}$ (Fig. 3c). Upon illumination with EP light, the maximum $V_{oc}$ values drop but remain alike for opposite helicity, i.e., for left-handed and right-handed EP light. The trend continues for the CP light as the $V_{oc}$ values remain independent of the helicity of the light and are around 20V. This is rather surprising for two reasons. Firstly, the substantial response measured in x-direction completely contradicts Eq. (2b), according to which response to CP light should be zero. And secondly, the response is independent of the light helicity.

To analyze this anomaly, we shift our focus to the individual CBPV responses arising from each of the two domain variants. The responses are summarized in Table 1 comprising Equation (3a)–(3d). Interestingly, the calculated current originating from the CBPV is perpendicular to corresponding polarization vector, i.e., for domain variant $P_L$ with polarization pointing in $[111]_{pc}$, RCP and LCP light leads to a photovoltaic current in $[1\bar{1}0]_{pc}$ and $[\bar{1}10]_{pc}$, respectively.

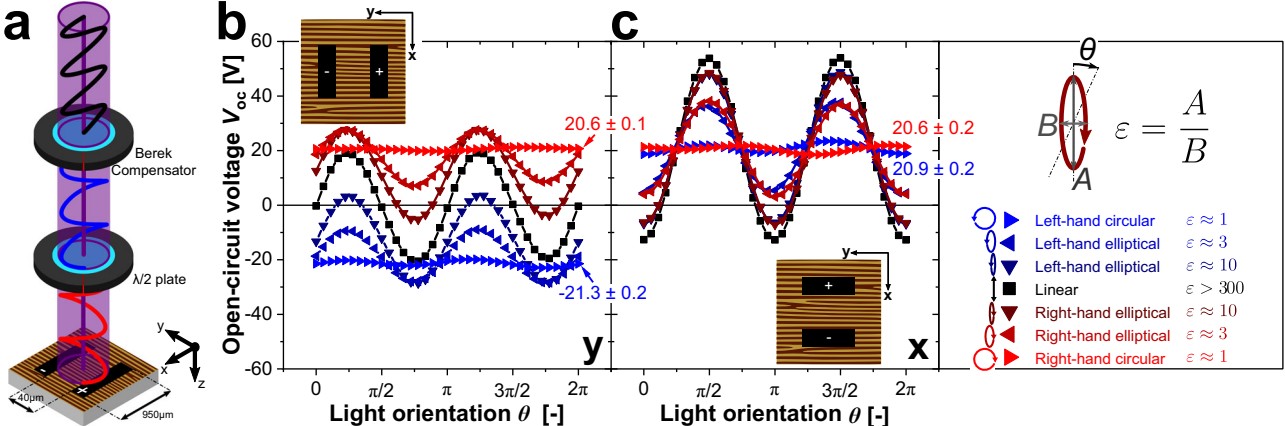

**Fig. 3 Deconvolution of the BPV response into linear and circular effects. a** Schematic of the extended setup (not to scale) comprising a Berek compensator (tunable waveplate) to set arbitrary polarization state of the light $\varepsilon$ with a subsequent $\frac{\lambda}{2}$ plate to rotate the orientation of the light polarization by $\theta$. $V_{oc}$ as a function of the light polarization orientation $\theta$ for different light polarization states measured in **b** y-direction and **c** x-direction. Right- and left-handedness of light is colored red and blue, respectively. The light polarization state as varied from linear ($\varepsilon > 300$) over two elliptical ($\varepsilon \approx 10$, $\varepsilon \approx 3$) to circular ($\varepsilon \approx 1$) light polarization.

Also, it is imperative to state that the tensorial assessment is based on a presumption that, both, RCP and LCP light interact equally with domains exhibiting different polarization variants $P_L$ and $P_R$. However, an alternative situation can be also presumed wherein RCP light interacts with only domains of one explicit domain variant, in this case $P_R$, while LCP light interacts with domains of the other variant. In this case, the photovoltaic response in y-direction will exhibit opposite polarity under RCP and LCP light, while, will be identical and non-zero in x-direction. This is in complete coherence with the experimental findings.

**Analysis of the CBPV and differential light interaction**. Based on the analytical assessment, we designed a measurement geometry to focus separately on the CBPV response from each variant under LCP and RCP light. Electrodes were engineered at an angle of ±45° with respect to the domain walls. In doing so, the electrodes are aligned parallel to the projection of the polarization vector of $P_L$ or $P_R$ in the xy-plane, while the in-plane projection of the other variant is orthogonal to the electrode. Keeping in perspective the calculated response given in Table 1, the expected $V_{oc}$ arising from variant $P_L$ and $P_R$ are schematically shown in Fig. 4a. For the −45° electrode geometry, the LCP light should provoke a considerable photovoltaic response, whereas with RCP the response should vanish. A reversed

behavior is expected in the +45° electrode geometry. This scenario is confirmed experimentally. It is amply evident that in the −45° configuration depicted in the Fig. 4b only LCP light results in a substantial $V_{oc}$ of 26.0V, while the RCP light induces a $V_{oc}$ of negligible magnitudes ( − 0.9V). In the +45° configuration, instead of LCP, the RCP results in a $V_{oc}$ of 25.0V and LCP induces a minimalistic $V_{oc}$ of −0.9V (Fig. 4c). The experimental findings agree with the scenario derived from the tensorial analysis in Table 1 and schematically demonstrated in Fig. 4a.

The agreement evidently suggests the need of differential interaction between CP light and domain variants. The interaction could be either involving absorption, i.e., circular dichroism (CD), or scattering. We conducted Raman scattering experiments similar to Raman optical activity (ROA) measurements. ROA is a known tool to investigate circular dichroic behavior[31]. Details about the Raman experiments can be found in the Supplementary Discussion and in the Methods section. Spatially-resolved Raman scattering experiment with CP light excitation in combination with principle component analysis (PCA) enabled the visualization of the ferroelectric domain arrangement, which was also confirmed by PFM measurements of the same region on the sample (Supplementary Fig. 8). Because of the usage of CP light, the observed changes in the Raman spectra, unlike other Raman scattering experiments (using LP light excitation)[32,33], cannot be explained by a different orientation of the light towards each domain variants. The change must originate from a differential interaction of CP light and the different domain variants and therefore, the Raman experiments further strengthen the presumed domain-specific light interaction.

However, it is imperative to mention that domain-selective light absorption, i.e. CD, cannot be ruled out and is a plausible scenario. As a matter of fact, differential circular dichroic behavior has been reported in poly-domain $BiFeO_3$ single crystal using (polarized) photo electron emission microscopy (PEEM)[34]. But the investigation of the dichroic behavior in domains with widths of only a few hundreds of nanometers is challenging because of the limited spatial resolution of typical characterization methods.

Nevertheless, the differential light–domain interactions (absorption or scattering) are an outcome of chirality. In the case of $BiFeO_3$, the crystal structure itself does not posses any chirality. However, a chiral order linked with the existence of a

**Table 1 CBPV contribution from each ferroelectric domain variant.**

|  | Right-handed circularly polarized light RCP ↻ |  | Left-handed circularly polarized light LCP ↺ |  |
|---|---|---|---|---|
| Domain variant $P_L$ | $j_i^{RCP,P_L} = \frac{I\beta_{12}^C}{2\sqrt{3}} \begin{pmatrix} +1 \\ -1 \\ 0 \end{pmatrix}$ | (3a) | $j_i^{LCP,P_L} = \frac{I\beta_{12}^C}{2\sqrt{3}} \begin{pmatrix} -1 \\ +1 \\ 0 \end{pmatrix}$ | (3b) |
| Domain variant $P_R$ | $j_i^{RCP,P_R} = \frac{I\beta_{12}^C}{2\sqrt{3}} \begin{pmatrix} -1 \\ -1 \\ 0 \end{pmatrix}$ | (3c) | $j_i^{LCP,P_R} = \frac{I\beta_{12}^C}{2\sqrt{3}} \begin{pmatrix} +1 \\ +1 \\ 0 \end{pmatrix}$ | (3d) |

Calculated CBPV response for the different domain variants $P_L$ and $P_R$ under illumination with RCP and LCP light, respectively. With respect to presumed the domain-specific light interaction, the CBPV current densities $j_i^{RCP,P_L}$ (Equation 3a) and $j_i^{LCP,P_R}$ (Equation 3d) vanish. The resulting CBPV response consists out of $j_i^{LCP,P_L}$ (Equation 3b) generated in $P_L$ under LCP light illumination and $j_i^{LCP,P_R}$ (Equation 3c) generated in $P_R$ under RCP light illumination. NOTE: Direction of $V_{oc}$ is antiparallel to the direction of $j_i$ (Compare Fig. 2b).

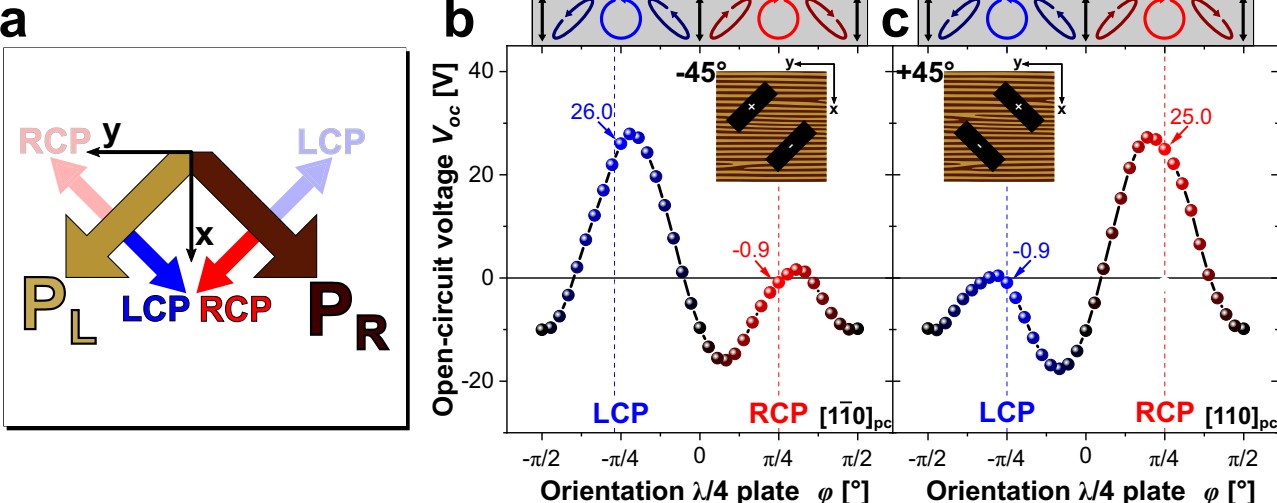

**Fig. 4 Electrode alignments to show dominance of LCP over RCP response and vice versa. a** Direction of $V_{oc}$ based on the proposed scenario above to describe the observed CBPV response (Equation 3b/3c in Table 1). The red and blue arrows implicate the direction of the $V_{oc}$ corresponding to RCP and LCP light, respectively. $V_{oc}$ as a function of $\frac{\lambda}{4}$ plate orientation $\phi$ measured with electrodes aligned along **b** $[1\bar{1}0]_{pc}$ (−45°). and **c** $[110]_{pc}$ (+45°).

cycloidal antiferromagnetic spin texture has been concluded from neutron diffraction measurements in $BiFeO_3$ ceramics[35]. This was also confirmed with neutron scattering[36] and hard X-ray magnetic scattering measurements[37] in single-crystalline $BiFeO_3$. Very recent studies conducted with nitrogen-vacancy magnetometry, and involving $BiFeO_3$ thin films, have explicitly revealed the alignment of antiferromagnetic order in the form of cycloid which in simple terms can be perceived as a periodic magnetic object with a chiral character. The spin cycloid propagates within individual and periodically arranged ferroelectric domains in $BiFeO_3/DyScO_3$ thin films[25,26,38]. The propagation vector of the cycloid remains perpendicular to the direction of ferroelectric polarization in each domain. Therefore, in a periodic domain pattern, of the kind utilized here, the propagation direction of the spin cycloid sequentially changes from one domain to the other by 90°. Furthermore, the configuration of the cycloid was found to be particularly sensitive to the strain in the film arising from the mismatch between the in-plane lattice parameters of the $BiFeO_3$ and substrates[26]. As a result, compressive strain of around 0.35% forces the propagation vector of cycloid to remain in the $(111)_{pc}$ plane, while under tensile strain the vector is restricted to $(1\bar{1}0)_{pc}$[24,26,39]. From the state of strain and domain arrangement in the $BiFeO_3/DyScO_3$ thin films analyzed in this study, the existence of spin cycloid can be postulated. Presuming a chirality flip associated with different cycloidal propagation direction, the observed behavior could arise from the differential interaction of CP light with the chiral magnetic texture, as has been also observed with soft resonant elastic X-ray scattering experiments[26,38]. Interestingly, a recent work has also demonstrated chiral arrangement of electric polarization within 71° domain walls[38]. Hence it becomes apparent that there are two chiral textures at play within the $BiFeO_3/DyScO_3$ samples under investigation in this work.

Furthermore, under higher compressive strain, the cycloid breaks and is replaced by G-type antiferromagnetic order[24,40]. A similar scenario is also encountered in $BiFeO_3$ films grown directly on $SrTiO_3$ $(001)_{pc}$ or $SrTiO_3$ $(111)_{pc}$ wherein the cycloidal order is absent[40–42]. Curiously, a periodic domain pattern is also missing in such samples which certainly hints towards its necessity for the manifestation of an ordered or harmonic spin cycloid arrangement. Precisely this aspect has been also observed with $BiFeO_3$ films grown on $DyScO_3$, albeit with mosaic domain pattern. Mössbauer spectroscopy was implemented to confirm that despite favorable epitaxial strain, a mosaic-like domain pattern is associated with an anharmonic cycloidal order due to significantly higher density of domain walls[43]. Therefore, it can be postulated that in such mosaic-like domain patterns the domain-specific light interaction, and anomalous CBPV effect, will be also substantially suppressed. We attempted to assess this condition with $BiFeO_3$ sample grown directly on $SrTiO_3$ and of similar thickness (Supplementary Fig. 9). The presence of four domain variants culminates in a mosaic-like domain pattern. A distinct LBPV response was measured and the results are in agreement with our previous study. However, the difference between the responses acquired with RCP and LCP light is around 2.5 V which is much less than measured in samples with stripe-like domain pattern (40V in Fig. 2c/3b).

**Discussion.** In this work we have demonstrated the photovoltaic response in $BiFeO_3$ under illumination with CP light. The periodic array of domains was critical for the observation of some rather compelling characteristics. Initially, as the polarization of the light was gradually varied from linear to elliptical to circular, the photovoltaic response was found to be notably higher for the elliptical polarization. The overall response was found to be a sum of contributions arising form linear and circular BPV effect. A modified setup allowed us to separate the resultant photovoltaic effect into its distinct contributions.

Analysis of the CBPV response with the associated tensors explicitly suggests the anomalous character of the BPV effect and the necessity of helicity-dependent interaction of light with domain variants. Recent studies have shown that films with periodic domain arrangement exhibit a magnetic texture formed by the regular arrangement of spin cycloid, orthogonally connected with the polarization vector of each domain. This chiral magnetic texture could be the origin of the differential light–domain interactions. As a matter of fact, analogous principles are also utilized to investigate the profile of the magnetic order and unambiguously differentiate between chiral and non-chiral arrangement[27]. However, the metallicity of the material itself restricts the usable wavelength to only X-rays in scattering mode. Other modes such as reflection and fluorescence have been also employed based on the insulating character of the material. In this regard, $BiFeO_3$ presents some apparent advantages. First, the bandgap of $BiFeO_3$ falls within visible range and a photon energy of 3.06 eV is sufficient for the BPV to arise, providing a direct evidence of light–matter interaction. Second, the periodic arrangement of domains in thin films essentially also assists in achieving a connected chiral antiferromagnetic order. The overlap of these two aspects presents a probable explanation for the observed effects.

## Methods

**Pulsed laser deposition (PLD) growth.** The $BiFeO_3$ thin films were grown on single-crystalline $DyScO_3$ $(110)_{orth}$ and $SrTiO_3$ $(001)_c$ substrates using a pulsed laser deposition system (SURFACE PLD-Workstation). During deposition, the substrate was kept at 625 °C and exposed to an oxygen partial pressure of 0.145 mbar. The distance between the stochiometric ceramic $BiFeO_3$ target and the substrate is set to 60 mm. The KrF excimer laser was set to energy densities in the range of $1.2-1.34\,J\,cm^{-2}$ with a pulse frequency of 2Hz.

**Piezo-response force microscopy (PFM).** PFM images were acquired with a Park NX10 system combined with an external lock-in amplifier (Zurich Instruments). The AC voltage (3 V, 20 kHz) was applied through a cantilever equipped with a platinum-coated tip (MikroMasch NSC 14).

**Device fabrication.** A conventional photolithography process has been used to structure the rectangular top electrode pairs ($950 \times 400\,\mu m^2$, 40 $\mu m$ spacing). A subsequent evaporation with platinum-palladium alloy (Pt:Pd 80:20) with a thickness of ~70 nm were achieved using a DC sputter machine.

**Photoelectrical measurements.** A high impedance electrometer (Keithley 6517B) acted as a voltage source (IV-characteristics, switching voltage) and simultaneously measured the current. The samples were illuminated by a diode laser (Cobolt 06 MLD) with a wavelength of 405 nm and 30 mW power.

**Raman spectroscopy.** Spatial-resolved Raman scattering experiments were recorded using a confocal Raman microscope setup (Renishaw, inVia) which was equipped with a 532 nm laser as excitation source, notch filter, a turnable grating (1800 lines $mm^{-1}$), and a CCD camera. Circular laser polarization was provided by inserting a quarter-wave plate into the excitation laser beam. A ×100 objective was used to focus the laser spot (1 µm) on the sample and to collect the scattered Raman light, respectively. A spatial resolution below the laser spot size was achieved by using the StreamLine$^{TM}$ high-resolution mode of the Raman instrument. In this mode, an increased spatial resolution is achieved by reducing the read-out area of the CCD detector during signal recording. Prior to the measurement, the instrument was calibrated to a band at 520.4 $cm^{-1}$ of a polycrystalline silicon disc. The sample was placed on a xy-stage (Renishaw) and an area of $7.6 \times 7.6\,\mu m^2$ was scanned using streamline-high resolution mode. The laser intensity was set to 5% (~1.5 mW) and spectra between 100 and 700 $cm^{-1}$ were recorded for 2 s per measurement point. Data were analyzed by a principle component analysis (PCA) with two components using the WiRE 3.4 software (Renishaw).

## Data availability
The data that support the findings of this study are available from the corresponding author upon reasonable request.

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

## Acknowledgements
We thank Prof. Kathrin Dörr and Dr. Diana Rata for the X-ray measurements, Marian Lisca for the technical support, Dr. Bodo Fuhrmann and Dipl.-Phys. Sven Schlenker for their support with the facilities at the Interdisziplinäre Zentrum für Materialwissenschaften (IZM). Financial support from Bundesministerium für Bildung und Forschung (BMBF) Project No. 03Z22HN12, Deutsche Forschungsgemeinschaft (DFG) within Sonderforschungsbereiche (SFB) 762 (project A12), and Europäischen Fonds für regionale Entwicklung (EFRE) Sachsen-Anhalt is gratefully acknowledged.

## Author contributions
A.B. and D.S.K. designed and conceived the experiments. D.S.K. was responsible for thin film growth, PFM analysis, and photoelectric measurements. Y.Y. and L.M. participated in structural characterization. M.S. performed the Raman scattering experiments. A.B. and D.S.K. co-wrote the manuscript with inputs from all the co-authors

## Funding

## Competing interests
The authors declare no competing interests.
