## [Peer Review File · Nature Communications]

REVIEWER COMMENTS

Reviewer #1 (Remarks to the Author):

Knoche et al. report on some very interesting and surprising results probing circular bulk photovoltaic effects in stripe domain BFO. The data is intriguing and the experiments are carefully carried out. In particular they make a highly surprising conclusion namely that the domains exhibit a strong circular dichroism. The results will certainly be of interest to the broad community interested in ferroelectrics and associated light-driven responses. I have a number of questions however regarding the interpretation of the experiments:

1. In Fig.2 - normally to isolate the CBPV response one would look at the difference signal LCP-RCP which cancels the linear contribution. As far as I can tell the authors don't do this. This would be helpful in examining the different contributions to the response.

2. They write "It must be emphasized here that the magnitude of photovoltaic current is substantially dependent on the intensity of light." Would be good to define 'substantially dependent' rather than make the reader go to the supplementary to understand what this means.

The conclusions on the CD response arise first from the surprising results of Fig. 2c which show a clear current induced along the $x=[010]$ direction. In contrast the calculation presented in the supplementary shows that the contribution along x cancels when including the contributions from both L and R domains. Thus the authors conclude that only one of the domains actually contributes. It is shown both through an analysis of the x -oriented electrodes and via measurements with electrodes at 45 that the experimental results are consistent with a CD response.

3. If there was a significant CD in BFO, would not this be observable via simple optical spectroscopy in say a mono domain film? This seems like a very simple test and more direct with respect to a CD response.

4. Also, the 45 deg electrode configuration is chosen so that the electrodes are parallel to P_R or P_L. Is not this angle actually $\arccos(1/\sqrt{3}) = 55$ deg (as shown in fig.1c e.g. the angle between P and $[100]_{pc}$). If so there should be a contribution at 45 deg.

5. The conclusions are based on an assumed crystallographic structure from when the BPV tensor is obtained. Can the authors rule out the fact that the structure around the domain walls is modified such that the anomalous response observed arises from the domain walls themselves rather than a bulk property of BFO? This is related to my question 1.

In summary the results reported here are intriguing, however I recommend the authors address the points above.

The authors report the observation of striking CBPE results in BiFeO₃ with well-distributed striped domains. Both LBPE and CBPE data are of high quality. The reviewer also appreciates the efforts on analyzing LBPE and CBPE currents (though most data are in voltage form) in the striped structures. However, there are some technical issues and theoretical analysis that have to be addressed before we move this manuscript forward. With updated technical analysis, the authors are requested to update the relevant technical discussions (e.g. some “surprising results” are indeed not surprising). Here are the technical comments to be addressed. I would like to see a revised manuscript and authors’ responses to my comments if the editor decides to move this manuscript forward.

- (1) Additional factors such as magnetism and magnetic microstructure have to be characterized to explain the observed CD phenomena. Unfortunately, current data do not adequately explain why CD is there. The current explanations are a collection of some published literature rather than correlating them to the observed phenomena. It is fine that the authors do not conduct quantitative studies. But they at least should provide qualitative explanation between CD and the CD-related structures/factors in their own BiFeO₃ system. For example, how magnetic structure modifies the CBPE and further yields CD should be presented. Experimental data to support this missing link are preferred.
- (2) The authors claim they are surprised by the observation of same-sign current (the authors most time use voltage for experiments but do theoretical analysis using currents so here I use current to represent voltage/current to eliminate confusion) along x direction under both left and right circularly polarized light (Fig. 3c). They claim these results are contradictory to equation 2. For me, it seems that this is not surprising. I am surprised by the observation of non-zero CBPE current/voltage along y direction rather than x direction. It seems the relevant derivation on equation 2 could be wrong due to the misuse of symmetry matrix. Here are my comments.

This is the so called “contradictory results” (see the figure below).

The so-called “contradictory results” from authors’ analysis.

I think the so called “contradictory results” come from the misuse of symmetry operation matrix (see the figure below): based on current domain structure, a reflection operation matrix should be used rather than rotation matrix (the two variants must be twin-like so you expect a mirror symmetry). If a reflection operation matrix is used in this derivation, then the authors should not see the contradictory results. In other words, the “contradictory results” are the ones along y-direction.

$$a_{P_R} = \begin{pmatrix} \cos(\frac{\pi}{2}) & \sin(\frac{\pi}{2}) & 0 \\ -\sin(\frac{\pi}{2}) & \cos(\frac{\pi}{2}) & 0 \\ 0 & 0 & 1 \end{pmatrix} a_{P_L} = \begin{pmatrix} 0 & 1 & 0 \\ -1 & 0 & 0 \\ 0 & 0 & 1 \end{pmatrix} a_{P_L} = \begin{pmatrix} -\frac{1}{\sqrt{2}} & -\frac{1}{\sqrt{6}} & \frac{1}{\sqrt{3}} \\ 0 & -\frac{2}{\sqrt{6}} & -\frac{1}{\sqrt{3}} \\ \frac{1}{\sqrt{2}} & -\frac{1}{\sqrt{6}} & \frac{1}{\sqrt{3}} \end{pmatrix} \quad (S12)$$

A reflection matrix should be used rather than rotation matrix.

Based on these arguments, the current equation 2 should not be right. So could the authors please update the derivations in supporting information and update equation 2?

Should not be zero based on tensor analysis

$$\text{LCP} \odot : j_i^{\text{LCP}} = \frac{I}{\sqrt{3}} \begin{pmatrix} 0 \\ -\beta_{12}^c \\ 0 \end{pmatrix} \quad (2a) \quad \text{RCP} \odot : j_i^{\text{RCP}} = \frac{I}{\sqrt{3}} \begin{pmatrix} 0 \\ \beta_{12}^c \\ 0 \end{pmatrix} \quad (2b)$$

Should be zero based on tensor analysis

- (3) The authors always use open circuit voltages for data collection but their analysis is all on currents. Please explain the reason.
- (4) The authors are suggested to sketch the incidence direction of illumination with respect to BFO in a 3D manner. The current drawing in Fig. 2a is bit confusing.
- (5) The conclusion of helicity-dependent absorption of light in domain variants is not proved. The authors only collect currents/voltages. Absorption data is needed.
- (6) This statement "Unraveled the first time an intricate relation between the magnetic texture and the bulk photovoltaic effect in BiFeO3" is not true. I do not see any experimental data to support this statement. Thus, experimental evidence and theoretical rigorousness are needed for supporting the relation between magnetism and CD.
- (7) In SI, S6 is identical to the one in the reference S1 but S7 is not. Why?
- (8) Please double check this equation and other similar ones. The second \mathbf{e} shall be in a complex conjugate form.

$$J_{i,P_R}^{\text{LCP}} = \frac{I}{2} a_{P_R} \beta_{ij}^c i [\vec{e}_{\text{LCP,h}} \times \vec{e}_{\text{LCP,h}}] = \frac{I}{2\sqrt{3}} \begin{pmatrix} -\beta_{12}^c \\ -\beta_{12}^c \\ 0 \end{pmatrix}$$

- (9) In line 100, you claim that '...a 1/2 plate positioned after the l/4 plate... the extracted LBPV contribution is shifted while the CBPV contribution remains unchanged'. Could you please clarify this result more? A 1/2 wave plate will switch the handedness of the light and you may expect a shifted CBPE?
- (10) To prove the existence of CD response in your samples, it would be more convincing if you simply conduct a test on a CD spectrometer. Ordinary CD spectrometer can collect signals from

UV to IR region, which perfectly covered the wavelength used in your test and can provide valuable evidence.

- (11) It would be helpful to estimate how much of PBV signal is canceled out by the coexistence of two domain orientations, similar to the analysis in Ref. 4.
- (12) What are the electrodes in Fig. 2a?
- (13) Line 213, there may be a typo there: "The distance between the stoichiometric The KrF excimer laser". The stoichiometric may be substrate?
- (14) Line 42, "analyzes" should be "analyze".
- (15) There is a small typo in line 93 and line 98. The circularity of the light should follow $\sin(2\varphi)$ rather than $\sin(\varphi)$.

Response to Reviewers: Circular Bulk Photovoltaic Effect in BiFeO₃ and Impact of Circular Dichroism

David S. Knoche, Matthias Steimecke, Yeseul Yun,
Lutz Mühlenbein, Akash Bhatnagar

October 23, 2020

**1 Reviewer 1**

***"Knoche et al. report on some very interesting and surprising results probing***
***circular bulk photovoltaic effects in stripe domain BFO. The data is intriguing and***
***the experiments are carefully carried out. In particular they make a highly surprising***
***conclusion namely that the domains exhibit a strong circular dichroism. The results***
***will certainly be of interest to the broad community interested in ferroelectrics and***
***associated light-driven responses. I have a number of questions however regarding***
***the interpretation of the experiments:"***

We thank the Reviewer for positively assessing our work and appreciate the con-
structive and helpful suggestions very much. In the following text, we have provided a
point-by-point response to the queries raised by the Reviewer.

Q1

"In Fig.2 - normally to isolate the CBPV response one would look at the difference
signal LCP-RCP which cancels the linear contribution. As far as I can tell the
authors don't do this. This would be helpful in examining the different contributions
to the response."

A1

We agree with the Reviewer that indeed, in scenarios wherein circular dichroism (CD)
is under investigation, a difference signal (LCP-RCP) is typically utilized. However, in
this study, the use of such a difference signal may result in rather misleading interpre-
tations. For instance, in the case of the open-circuit voltages (V_{oc}) measured along the
orthogonal directions, the difference signal will result in values of 0.3 V (Fig. 3c) and
30 -41.9 V (Fig. 3b). These values do not depict the impact of circular dichroism because
the measured photovoltaic response is influenced by the symmetry relations used in the
tensorial analysis AND the presumed circular dichroism.

Figure R1: **a** Schematic of the light polarization characterization setup. The set light polarization is rotated using the subsequent half-wave plate and the laser power for the major (A) and minor (B) axis is measured using a silicon photo detector (Thorlabs, DET100A2). **b** Ellipticity ratio ϵ as a function of the tilt angle of the Berek Compensator. Depending on the tilt angle, the pathway of the light through the birefringent MgF_2 plate is changed. This changes the resulting retardance and thus the outcoming light polarization from linear over elliptical to circular.

Nevertheless, to be certain of minimalist contributions arising from linear polarization,
we conducted an in-depth analysis of the state of light polarization which is presented
in Figure R1. The ellipticity ratio ϵ was plotted as a function of the tilt of angle of
Berek compensator. ϵ is in excess of 300 for linearly polarized light and ~ 1.1 for circular
polarization. In conjunction, the values are overlapping for the right- and left-hand
circularly polarized light which highlights the precision of the setup. As a matter of fact,
the small deviation from an ideal CP state ($\text{ER}=1$) is also visible in the photovoltaic

response. The waviness of the response acquired with RCP and LCP light shown in
Fig 3 b,c of the manuscript can be attributed to the deviation.

**Action taken:** We added Fig. R1 to the Supplementary material (Fig. S7).

Q2

*"They write "It must be emphasized here that the magnitude of photovoltaic cur-*
*rent is substantially dependent on the intensity o light." Would be good to define*
*'substantially dependent' rather than make the reader go to the supplementary to*
*understand what this means."*

A2

**Action taken :** We agree and have accordingly changed the sentence in the manuscript:
"It must be emphasized here that the magnitude of photovoltaic current **scales up**
**linearly with** ~~is substantially dependent on~~ the intensity of light."

Q3

*"The conclusions on the CD response arise first from the surprising results of Fig.*
*2c which show a clear current induced along the $x=[010]$ direction. In contrast*
*the calculation presented in the supplementary shows that the contribution along*
*x cancels when including the contributions from both L and R domains. Thus the*
*authors conclude that only one of the domains actually contributes. It is shown*
*both through an analysis of the x -oriented electrodes and via measurements with*
*electrodes at 45 that the experimental results are consistent with a CD response.*

*If there was a significant CD in BFO, would not this be observable via simple*
*optical spectroscopy in say a mono domain film? This seems like a very simple*
*test and more direct with respect to a CD response."*

A3

The mono-domain state in BiFeO₃ can be stabilized only under the influence of a larger
epitaxial strain imposed by the substrates. In this regard, growth of BiFeO₃ on (111)-
(1) and (001)-oriented (2, 3) SrTiO₃ substrates may result in dominant crystallization
in mono-domain state. However, the resultant polarization state (especially BiFeO₃ on
(111)) is different and the large compressive strain has been demonstrated to be rather
detrimental for the spin cycloidal arrangement and instead promotes the formation of
collinear G-type antiferromagnetic structure.(4) Therefore, it can be proposed here that

a mono-domain state does not provide a favorable scenario for the observation of circular
dichroism.

Interestingly, circular dichroism has been observed in BiFeO₃ single crystals exhibiting
ferroelectric domains with sizes >30 μm.⁽⁵⁾ Photoemission electron microscopy (PEEM)
setup was used in conjunction with a near-UV laser light source with a photon energy of
around 4 eV. Circular dichroic effects were observed from neighboring domains separated
by a domain wall, likewise the dichroic behavior we propose. The inherent magnetic
structure was only tentatively suggested as a possible origin for the observed dichroism.

A critical restriction in the observation of opposite circular dichroism arising from
narrowly spaced domains (like in this study) is the spatial resolution of the characteri-
zation technique. We performed Raman scattering experiments under circular polarized
light with a small spot size of the excitation laser light (~1 μm). To further improve the
spatial resolution, a special mode of the Raman instrument was enabled, which allowed
to analyze the center of the scattered Raman signal (more details, Reviewer 2 Q5).

Q4

***"Also, the 45 deg electrode configuration is chosen so that the electrodes are***
***parallel to P_R or P_L . Is not this angle actually $\arccos(1/\sqrt{3}) = 55$ deg (as shown in***
***fig.1c e.g. the angle between P and $[100]_{pc}$). If so there should be a contribution***
***at 45 deg."***

A4

We acknowledge the inaccurate description of the electrode geometry and thank the
Reviewer for the indication. In all cases presented in this study, the electrodes were
evaporated on top of the (001)_{pc} BiFeO₃ surface, e.g. in the **xy**-plane. The light propa-
gates perpendicular to this plane along the **z**-direction. Consequently, the angle between
the light propagation direction and the polarization direction is, in each domain variant,
$\arccos(1/\sqrt{3}) = 54.7^\circ$. This aspect is already incorporated in the tensorial calculation.
For the all the considered geometries, the CBPV response is zero along the **z**-direction,
and the generated photocurrents are constrained within the **xy**-plane.

In case of the $\pm 45^\circ$ electrode configuration, the electrodes do NOT run parallel to the
polarization direction P_R or P_L , but instead are parallel to the planar projection of P_R
or P_L in the **xy**-plane. The schematic in Fig. 4a of the manuscript depicts the resultant
direction of the generated V_{oc} in relation to the planar projection of the polarization in
the **xy**-plane.

**Action taken:** The descriptions of the electrode geometries are clarified throughout
the manuscript.

**Q5**

***"The conclusions are based on an assumed crystallographic structure from when***
***the BPV tensor is obtained. Can the authors rule out the fact that the structure***
***around the domain walls is modified such that the anomalous response observed***
***arises from the domain walls themselves rather than a bulk property of BFO? This***
***is related to my question 1"***

**A5**

Ever since the discovery of the photovoltaic effect in BiFeO₃ (6, 7), the role of the
domain walls has been extensively deliberated upon because of their outstanding and
different properties compared to the bulk (8, 9).

We agree with you, that there might be a role of the modified crystal structure at
the domain walls. In a very recent study about the antiferromagnetic spin cycloid
in BiFeO₃/DyScO₃ heterostructures, Chauleau *et al.* attributed the circular dichroic
behavior observed in resonant elastic x-ray scattering (REXS) measurements partially
to the presence of a chiral arrangement of the ferroelectric polarization within the domain
walls.(10) Additionally, they observed another circular dichroic behavior originated from
the chiral antiferromagnetic texture (non-collinear spin cycloid) within the ferroelectric
domain.

Because of the possible simultaneous existence of both electric (within the domain
walls) and magnetic (within the bulk) chirality, it is not possible to clearly attribute
our anomalous response to either one of them. However, due to the simple fact of a
larger volume fraction of the domain compared to the domain wall, we assume that this
anomalous response is attributed to the domain and thus to the magnetic chirality.

Also, noteworthy to mention is the nano-scopic studies conducted with a tip of atomic
force microscope which illustrated the photoconductive character of the domain walls
rather than the previously assumed role in charge separation.(11)

**Action taken:** We added the information about the existence of chirality of the
ferroelectric polarization within the domain wall to our discussion.

_____

2 Reviewer 2

"The authors report the observation of striking CBPE results in BiFeO₃ with well-distributed striped domains. Both LBPE and CBPE data are of high quality. The Reviewer also appreciates the efforts on analyzing LBPE and CBPE currents (though most data are in voltage form) in the striped structures. However, there are some technical issues and theoretical analysis that have to be addressed before we move this manuscript forward. With updated technical analysis, the authors are requested to update the relevant technical discussions (e.g. some "surprising results" are indeed not surprising). Here are the technical comments to be addressed. I would like to see a revised manuscript and authors' responses to my comments if the editor decides to move this manuscript forward."

We greatly appreciate the assessment from the Reviewer and the constructive feedback. Below is a point-by-point response to all the issues and suggestions that were communicated.

Q1

"Additional factors such as magnetism and magnetic microstructure have to be characterized to explain the observed CD phenomena. Unfortunately, current data do not adequately explain why CD is there. The current explanations are a collection of some published literature rather than correlating them to the observed phenomena. It is fine that the authors do not conduct quantitative studies. But they at least should provide qualitative explanation between CD and the CD-related structures/factors in their own BiFeO₃ system. For example, how magnetic structure modifies the CBPE and further yields CD should be presented. Experimental data to support this missing link are preferred."

A1

Circular dichroism, i.e. differential absorption of circularly polarized light within a medium, is an outcome of chirality.⁽¹²⁾ In the case of BiFeO₃, the structure itself does not possess any chirality. However, thin film growth on asymmetric DyScO₃ substrates provide a lucrative platform to achieve long range arrangement of ferroelastic domains separated by 71° domain walls. The recent studies on BiFeO₃ thin films (mostly grown on DyScO₃) with nitrogen vacancy technique have explicitly revealed the alignment of AFM order in the form of cycloid which in simple terms can be understood as a periodic magnetic object with a chiral character.^(10, 13, 14) The interaction of CP with the chiral magnetic order can be considered as the one of the possible origins for circular dichroism. In addition, a recent work has also demonstrated chiral arrangement of electric polarization within domain walls.⁽¹⁰⁾ Hence it becomes apparent that there are

176 two chiral textures at play within the BiFeO₃ films under investigation in this work.
 Further investigations will be needed to de-convolute their respective contribution to
 the eventual circular dichroism.

We would like to reiterate here that the primary objective of this work remains the
 observation of circular photovoltaic effect and role of circular dichroism. In this context,
 and as per the suggestion of the Reviewer, we have now provided another hint of the man-
 ifestation of circular dichroism using laterally-resolved Raman scattering experiments.
 Further information is available in our response to question Reviewer 2 Q5.

**Action taken:** We changed the discussion concerning the influence of the magnetic
 structure on optoelectronic processes in materials in general and specified the discussion
 for BiFeO₃.

_____

Q2

***"The authors claim they are surprised by the observation of same-sign current (the***
 ***authors most time use voltage for experiments but do theoretical analysis using***
 ***currents so here I use current to represent voltage/current to eliminate confusion)***
 ***along x direction under both left and right circularly polarized light (Fig. 3c). They***
 ***claim these results are contradictory to equation 2. For me, it seems that this is not***
 ***surprising. I am surprised by the observation of non-zero CBPE current/voltage***
 ***along y direction rather than x direction. It seems the relevant derivation on***
 ***equation 2 could be wrong due to the misuse of symmetry matrix. Here are my***
 ***comments***

This is the so called "contradictory results" (see figure below).

The so-called "contradictory results" from authors' analysis.

***I think the so called "contradictory results" come from the misuse of symmetry***
 ***operation matrix (see the figure below): based on current domain structure, a***
 ***reflection operation matrix should be used rather than rotation matrix (the two***

**variants must be twin-like so you expect a mirror symmetry). If a reflection op-**
 **eration matrix is used in this derivation, then the authors should not see the**
 **contradictory results. In other words, the “contradictory results” are the ones**
along y-direction.

$$a_{P_R} = \begin{pmatrix} \cos(\frac{\pi}{2}) & \sin(\frac{\pi}{2}) & 0 \\ -\sin(\frac{\pi}{2}) & \cos(\frac{\pi}{2}) & 0 \\ 0 & 0 & 1 \end{pmatrix} a_{P_L} = \begin{pmatrix} 0 & 1 & 0 \\ -1 & 0 & 0 \\ 0 & 0 & 1 \end{pmatrix} a_{P_L} = \begin{pmatrix} -\frac{1}{\sqrt{2}} & -\frac{1}{\sqrt{6}} & \frac{1}{\sqrt{3}} \\ 0 & -\frac{2}{\sqrt{6}} & -\frac{1}{\sqrt{3}} \\ \frac{1}{\sqrt{2}} & -\frac{1}{\sqrt{6}} & \frac{1}{\sqrt{3}} \end{pmatrix} \quad (S12)$$

A reflection matrix should be used rather than rotation matrix.

**Based on these arguments, the current equation 2 should not be right. So could**
 **the authors please update the derivations in supporting information and update**
equation 2??"

$$\text{LCP}\odot: j_i^{\text{LCP}} = \frac{I}{\sqrt{3}} \begin{pmatrix} 0 \\ -\beta_{12}^c \\ 0 \end{pmatrix} \quad (2a) \quad \text{RCP}\odot: j_i^{\text{RCP}} = \frac{I}{\sqrt{3}} \begin{pmatrix} 0 \\ \beta_{12}^c \\ 0 \end{pmatrix} \quad (2b)$$

Should not be zero based on tensor analysis

Should be zero based on tensor analysis

A2

The application of a reflection matrix, instead of a rotation matrix, apparently will have
 an impact on the results of the tensorial analysis.

Our analysis is largely based on the work from Young *et al.*(15) wherein also a rotation
 matrix was used to realize transformation between domain variants separated by 71°
 domain walls. For convenience, in Fig R2 a schematic is presented which broadly depicts
 the comparison between our work and Young *et al.*(15) with regard to the coordinate
 systems and relation with the domain configurations. Due to the evident differences
 in the choice of conventions, a different transformation matrix was calculated in our
 analysis. This apparently also explains the slight differences in the matrices of the
 resultant domains variants, which are also enlisted in Tab. R1.

[Redacted]

Figure R2: Left (Adapted from Young *et al.*(15)): In order to changed from domain configuration R to L a 90° clockwise rotation around the y-axis can be used. The light propagation direction is along -y-direction Right (Our work): To change from domain configuration P_L to P_R a counter-clockwise rotation around the z-axis can be used. The light propagation direction is along +z-direction.

Table R1: Transformation matrices used in the work of Young *et al.*(15) (top) and in our work (bottom). In both cases, a rotation matrix is used to change the transformation matrix from one domain to the other in a 71° domain wall arrangement.

	Polarization variant 1	→	Polarization variant 2
Young et al. (15)	$R \parallel \begin{pmatrix} -1 \\ -1 \\ 1 \end{pmatrix}$ $\begin{pmatrix} \frac{1}{\sqrt{2}} & -\frac{1}{\sqrt{6}} & -\frac{1}{\sqrt{3}} \\ 0 & \frac{\sqrt{2}}{\sqrt{3}} & -\frac{1}{\sqrt{3}} \\ \frac{1}{\sqrt{2}} & \frac{1}{\sqrt{6}} & \frac{1}{\sqrt{3}} \end{pmatrix}$	90° CW y $\begin{pmatrix} 0 & 0 & 1 \\ 0 & 1 & 0 \\ -1 & 0 & 0 \end{pmatrix}$	$L \parallel \begin{pmatrix} 1 \\ -1 \\ 1 \end{pmatrix}$ $\begin{pmatrix} \frac{1}{\sqrt{2}} & \frac{1}{\sqrt{6}} & \frac{1}{\sqrt{3}} \\ 0 & \frac{\sqrt{2}}{\sqrt{3}} & -\frac{1}{\sqrt{3}} \\ -\frac{1}{\sqrt{2}} & \frac{1}{\sqrt{6}} & \frac{1}{\sqrt{3}} \end{pmatrix}$
Our work	$P_L \parallel \begin{pmatrix} 1 \\ 1 \\ 1 \end{pmatrix}$ $\begin{pmatrix} 0 & \frac{2}{\sqrt{6}} & \frac{1}{\sqrt{3}} \\ -\frac{1}{\sqrt{2}} & -\frac{1}{\sqrt{6}} & \frac{1}{\sqrt{3}} \\ \frac{1}{\sqrt{2}} & -\frac{1}{\sqrt{6}} & \frac{1}{\sqrt{3}} \end{pmatrix}$	90° CCW z $\begin{pmatrix} 0 & 1 & 0 \\ -1 & 0 & 0 \\ 0 & 0 & 1 \end{pmatrix}$	$P_R \parallel \begin{pmatrix} 1 \\ -1 \\ 1 \end{pmatrix}$ $\begin{pmatrix} -\frac{1}{\sqrt{2}} & -\frac{1}{\sqrt{6}} & \frac{1}{\sqrt{3}} \\ 0 & -\frac{2}{\sqrt{6}} & -\frac{1}{\sqrt{3}} \\ \frac{1}{\sqrt{2}} & -\frac{1}{\sqrt{6}} & \frac{1}{\sqrt{3}} \end{pmatrix}$

Nevertheless, as per the suggestion of the Reviewer, we have now also performed
 an analysis with a reflection operation matrix. The results, and comparison with the
 rotation matrix, are presented in Table R2. As indicated by the Reviewer, the response
 indeed vanishes in y-direction with a reflection matrix. However, the non-zero response
 in x-direction exhibits a change of sign with light chilarity, which apparently is not in
 agreement with our experimental findings.

Table R2: Calculated CBPV response for both light chiralities using a reflection transformation matrix (red, top) and a rotation transformation matrix (green, bottom) to describe the striped domain arrangement.

	RCP \odot	LCP \circ
Reflection	$j_i^{\text{RCP}^*} = \frac{I}{\sqrt{3}} \begin{pmatrix} \beta_{12}^{\text{C}} \\ 0 \\ 0 \end{pmatrix} \quad (\text{R.2a})$	$j_i^{\text{LCP}^*} = \frac{I}{\sqrt{3}} \begin{pmatrix} -\beta_{12}^{\text{C}} \\ 0 \\ 0 \end{pmatrix} \quad (\text{R.2b})$
Rotation	$j_i^{\text{RCP}} = \frac{I}{\sqrt{3}} \begin{pmatrix} 0 \\ -\beta_{12}^{\text{C}} \\ 0 \end{pmatrix} \quad (\text{R.2c})$	$j_i^{\text{LCP}} = \frac{I}{\sqrt{3}} \begin{pmatrix} 0 \\ \beta_{12}^{\text{C}} \\ 0 \end{pmatrix} \quad (\text{R.2d})$

Furthermore, we tested the validity of the reflection matrix by calculating the response
 for LBPV effect, which is presented in Tab. R3. In the resultant response, the pho-
 tocurrent completely loses the angular dependency, which is in contradiction to the
 previously published studies and the experimental findings of the present work (Fig. 2
 in the manuscript). We therefore are in a position to emphasize on the suitability of the
 rotation operation matrix instead of the reflection.

Table R3: Calculated LBPV respons using a reflection transformation matrix (red, top) and a rotation transformation matrix (grey, bottom) to describe the striped domain arrangement.

Linearly polarized light	
Reflection $\begin{pmatrix} 1 & 0 & 0 \\ 0 & -1 & 0 \\ 0 & 0 & 1 \end{pmatrix}$	$\begin{pmatrix} \left(\frac{\beta_{33}^L}{3\sqrt{3}} + \frac{2\beta_{31}^L}{3\sqrt{3}} + \frac{\beta_{22}^L}{3\sqrt{6}} + \frac{\beta_{15}^L}{6\sqrt{3}} \right) \\ \left(\frac{\beta_{33}^L}{3\sqrt{3}} - \frac{\beta_{31}^L}{3\sqrt{3}} - \frac{2\beta_{22}^L}{3\sqrt{6}} + \frac{\beta_{15}^L}{6\sqrt{3}} \right) \sin(2\theta) - \left(\frac{\beta_{22}^L}{\sqrt{6}} + \frac{\beta_{15}^L}{2\sqrt{3}} \right) \cos(2\theta) \\ \left(-\frac{2\beta_{22}^L}{3\sqrt{6}} - \frac{\beta_{15}^L}{3\sqrt{3}} + \frac{\beta_{33}^L}{3\sqrt{3}} + \frac{2\beta_{31}^L}{3\sqrt{3}} \right) \end{pmatrix} \quad (\text{R.3a})$
Rotation $\begin{pmatrix} 0 & 1 & 0 \\ -1 & 0 & 0 \\ 0 & 0 & 1 \end{pmatrix}$	$j_i^L = I \begin{pmatrix} \left(\frac{\beta_{33}^L}{3\sqrt{3}} + \frac{2\beta_{31}^L}{3\sqrt{3}} + \frac{\beta_{22}^L}{3\sqrt{6}} + \frac{\beta_{15}^L}{6\sqrt{3}} \right) + \left(\frac{\beta_{22}^L}{\sqrt{6}} + \frac{\beta_{15}^L}{2\sqrt{3}} \right) \cos(2\theta) \\ \left(\frac{\beta_{33}^L}{3\sqrt{3}} - \frac{\beta_{31}^L}{3\sqrt{3}} - \frac{2\beta_{22}^L}{3\sqrt{6}} + \frac{\beta_{15}^L}{6\sqrt{3}} \right) \sin(2\theta) \\ \left(\frac{2\beta_{22}^L}{3\sqrt{6}} + \frac{\beta_{15}^L}{3\sqrt{3}} - \frac{\beta_{33}^L}{3\sqrt{3}} - \frac{2\beta_{31}^L}{3\sqrt{3}} \right) \end{pmatrix} \quad (\text{R.3b})$

**Q3**

*"The authors always use open circuit voltages for data collection but their analysis*
 *is all on currents. Please explain the reason."*

**A3**

As per our understanding, this issues was already addressed in the manuscript (line
 80-83), wherein in we provide the reason for the use of open-circuit voltages instead of
 the short-circuit. In the home-built measurement setup, the optical parts (mostly the
 Berek compesnsator) can sometimes trigger slight changes in the light intensity, to which
 the photocurrent was found to be rather sensitive see Fig. S4. On the contrary, the V_{oc}
 appears to be largely independent of the light intensity above a certain threshold.

A set of representative data comprising of current (density)- voltage characteristics
 with different light orientation for linear light polarization are parallel to each other
 (Fig. R3a). Therefore, both the photovoltaic current (density) or open-circuit voltage
 can be used to describe the BPV response (Fig. R3b).

Figure R3: Description see paragraph above.

**Action taken:** We have now provided a more detailed explanation for the use of
 the V_{oc} over the current densities. Additionally, we added Fig R3 to the Supplemental
 Information (Fig. S5).

Q4

*"The authors are suggested to sketch the incidence direction of illumination with
 respect to BFO in a 3D manner. The current drawing in Fig. 2a is bit confusing."*

A4

**Action taken:** We followed the Reviewer's suggestion and updated the schematics in
 Fig. 2a and Fig. 3a in the new manuscript.

Q5

*"The conclusion of helicity-dependent absorption of light in domain variants is not
 proved. The authors only collect currents/voltages. Absorption data is needed."*

A5

Our answer to this question is also related to Reviewer 1 Q3 and Reviewer 2 Q10.

There are several reason, why it was not possible for us to perform absorption mea-
 surements on our thin films. Mostly, all of the difficulties are related to the fact that
 a domain configuration with domain widths in the range of only several hundreds of
 nanometer are present in our films.

To explain the unexpected CBPV response in our films, we propose the existence of
opposite circular dichroism for the different polarization variants. Consequently, CD
spectra for each polarization variant is needed. To the best of our knowledge, commer-
cially available CD spectrometer (mostly equipped with a Xe lamp) are not capable of
resolving structure in the submicrometer range.

As already discussed in Reviewer 1 A3, circular dichroism has been observed using
laser-based threshold photoemission electron microscopy (PEEM) in a BiFeO₃ single
crystal with multi-domain state.⁽⁵⁾ Having this observation in mind, we conceived a
Raman scattering experiment following partially the basic principle of Raman Optical
Activity (ROA) in collaboration with our colleagues. The Raman spectra were acquired
using a confocal Raman microscope setup (Renishaw, InVia). The sample was place
on a **xy**-stage providing a 100 nm positioning resolution. The polarization state of the
excitation laser light (532 nm) is tuned from linear to circular by inserting a quarter-
wave plate. In order to gain information from a very localized region of the sample, we
focused the laser beam to a spot size of $\sim 1 \mu\text{m}$. A spatial resolution below the laser spot
size was achieved by using the StreamLineTM high-resolution mode of the Raman instru-
ment. In this mode, an increased spatial resolution is achieved by reducing the read-out
area of the CCD detector during signal recording. A comparison between two spectra
observed under illumination with linear (polarization along **x**-direction) and circular
light polarization can be found in Fig. R4(a). In both spectra, three dominant modes
at 138 cm^{-1} , 174 cm^{-1} and 222 cm^{-1} were observed. This is in good agreement with
previous studies on BiFeO₃/DyScO₃ thin films.⁽¹⁶⁾ Contrarily, Talkenberger *et al.* used
a 442 nm linearly polarized excitation laser. Because of the smaller penetration depth
of the light ($\sim 75 \text{ nm}$), they did not separate modes at higher wave numbers due to the
DyScO₃ substrate (compare spectra of DyScO₃ substrate at bottom in Fig. R4(a)). Be-
cause of this contribution from the substrate, we were not able to observe BiFeO₃ modes
at higher wave numbers. In another study, Himcinshi *et al.* used Raman spectroscopy to
identify domains with different polarization variants in BiFeO₃ a single crystal.⁽¹⁷⁾ The
change in the Raman signal (peak area of 172 cm^{-1} mode) due to a different orientation
of the linear polarization of the excitation laser towards the corresponding polarization
direction was used to map a domain configuration with micrometer-resolution. When
performing the Raman scattering experiments, we observed a change of the ratio be-
tween the peak intensity of the 138 cm^{-1} and 174 cm^{-1} mode, however, under excitation
with circularly polarized light. The intensity changes were not as high as shown by
Himcinshi *et al.*⁽¹⁷⁾, which might also be related to the smaller domain size and the
different excitation wavelength.

Figure R4: **a** Normalized Raman spectra of $\text{BiFeO}_3/\text{DyScO}_3$ thin film under circular (red) and linear (black) laser light excitation. Bottom: Qualitative Raman spectra of a DyScO_3 substrate. Inset: Both PCA-component used to analyze the spatial-resolved Raman spectra. **b** Topography and **c** in-plane PFM phase image of pre-characterized region ($10 \times 10 \mu\text{m}^2$). The red squares indicates the position of the overlapped map acquired from the spatial-resolved Raman scattering experiments. **d** Normalized PCA-component II map ($7.6 \times 7.6 \mu\text{m}^2$) acquired from the spatial-resolved Raman scattering experiments overlapping the schematic domain configuration extracted from PFM image in **c**. The outline of the map is inserted in **b** and **c** (red square).

To systematically investigate, this position-dependent change in the Raman spec-
 tra, we characterized a $10 \times 10 \mu\text{m}^2$ region on our sample using piezo-response force mi-
 croscopy (PFM). The AFM image (Fig. R4(b)) reveals a topography with a similar
 roughness to the one shows in Fig.1(a) in our manuscript. The in-plane PFM (IP-
 PFM) phase image (Fig. R4(c)) show great similarities to Fig.1(b), however, we want
 to mention the larger domain width in this region ($\sim 250 \text{ nm}$ compared to $\sim 150 \text{ nm}$ in
 Fig.1(b)).

Then, we performed spatial-resolved Raman scattering experiments to acquire a two-
 dimensional grid of Raman spectra in a $8 \times 8 \mu\text{m}^2$ region in the vicinity of the PFM
 pre-characterized region. The Raman data (~ 6400 spectra) were processed using prin-
 ciple component analysis (PCA), a commonly used tool show data variance. The PCA-
 analysis divides the Raman spectra into two main components (see inset Fig. R4). The
 change of the 174 cm^{-1} and 138 cm^{-1} peak intensity ratio can be represented by the PCA-

component II. As a result, the position-dependent PCA-component can be mapped. To
compare two characterization techniques, the resultant PCA-component map is over-
lapped with the IP-PFM phase image Fig. R4(d). To match the shape of both pattern,
the PCA-component map is resized ($8 \times 8 \mu\text{m}^2 \rightarrow 7.6 \times 7.6 \mu\text{m}^2$). The lowest spatial res-
olution step of **xy**-positioning stage of the Raman microscope was used and a systemic
positioning error while working at the lowest resolution limit could explain the neces-
sity to resize the map to achieve a perfect overlap. However, the spatial resolution of
this experiment is limited to $\sim \lambda/2$. For thinner structures, a mixed Raman spectra
is obtained and it is not possible to clearly distinguish the different domains (compare
Fig. R4d upper and lower areas).

The possibility to resolve the ferroelectric domain arrangement using Raman scatter-
ing experiments under excitation with CP light suggests differential interaction between
CP light and the domains exhibiting different polarization variants.

**Action taken:** We added Fig. R4 to Supplementary information (Fig. S8) and added
a paragraph about the Raman experiments to the main text.

Q6

***"This statement "Unraveled the first time an intricate relation between the mag-
netic texture and the bulk photovoltaic effect in BiFeO3" is not true. I do not see
any experimental data to support this statement. Thus, experimental evidence and
theoretical rigorosness are needed for supporting the relation between magnetism
and CD."***

A6

We agree that the statement is not fully corroborated by the experimental findings.

**Action taken:** We corrected the statement to "Our findings ~~imply have unraveled for~~
~~the first time~~ an intricate relation between the magnetic texture and the bulk photo-
voltaic effect in BiFeO₃ thin films" in the new manuscript.

Q7

***"In S1, S6 is identical to the one in the reference S1 but S7 is not. Why?"***

A7

The differences in the shape of the 3rd rank tensor is due to the usage of a different
convention. In reference (18), the different layers of the tensor can be found in different
rows (general form of a third-rank tensor for point group 3m):

$$\begin{array}{l}
\text{1st layer } i = 1 \\
\text{2nd layer } i = 2 \\
\text{3rd layer } i = 3
\end{array}
\left(\begin{array}{ccccccccc}
0 & -T_{222} & T_{223} & -T_{222} & 0 & 0 & T_{131} & 0 & 0 \\
-T_{222} & 0 & 0 & 0 & T_{222} & T_{223} & 0 & T_{131} & 0 \\
T_{311} & 0 & 0 & 0 & T_{311} & 0 & 0 & 0 & T_{333}
\end{array} \right) \quad (\text{R.4})$$

↓

$$\begin{array}{ccc}
\text{1st layer} & \text{2nd layer} & \text{3rd layer} \\
i = 1 & i = 2 & i = 3
\end{array}
\left(\begin{array}{ccc|ccc|ccc}
0 & -T_{222} & T_{223} & -T_{222} & 0 & 0 & T_{311} & 0 & 0 \\
-T_{222} & 0 & 0 & 0 & T_{222} & T_{223} & 0 & T_{311} & 0 \\
T_{131} & 0 & 0 & 0 & T_{131} & 0 & 0 & 0 & T_{333}
\end{array} \right) \quad (\text{R.5})$$

We prefer to use the convention shown above, in which there is a block-wise layering of the third-rank tensor following reference (19).

The transformation of the LBPV (symmetric) is:

$$\text{Reference (18)} \left(\begin{array}{ccccccccc}
0 & -\beta_{222}^S & \beta_{131}^S & -\beta_{222}^S & 0 & 0 & \beta_{131}^S & 0 & 0 \\
-\beta_{222}^S & 0 & 0 & 0 & \beta_{222}^S & \beta_{131}^S & 0 & \beta_{131}^S & 0 \\
\beta_{311}^S & 0 & 0 & 0 & \beta_{311}^S & 0 & 0 & 0 & \beta_{333}^S
\end{array} \right) \quad (\text{R.6})$$

↓

$$\text{Our work} \left(\begin{array}{ccc|ccc|ccc}
0 & -\beta_{222}^L & \beta_{131}^L & -\beta_{222}^L & 0 & 0 & \beta_{311}^L & 0 & 0 \\
-\beta_{222}^L & 0 & 0 & 0 & \beta_{222}^L & \beta_{131}^L & 0 & \beta_{311}^L & 0 \\
\beta_{131}^L & 0 & 0 & 0 & \beta_{131}^L & 0 & 0 & 0 & \beta_{333}^L
\end{array} \right) \quad (\text{R.7})$$

The transformation of the CBPV (antisymmetric) is:

$$\text{Reference (18)} \left(\begin{array}{ccccccccc}
0 & 0 & -\beta_{131}^A & 0 & 0 & 0 & \beta_{131}^A & 0 & 0 \\
0 & 0 & 0 & 0 & 0 & -\beta_{131}^A & 0 & \beta_{131}^A & 0 \\
0 & 0 & 0 & 0 & 0 & 0 & 0 & 0 & 0
\end{array} \right) \quad (\text{R.8})$$

↓

$$\text{Our work} \left(\begin{array}{ccc|ccc|ccc}
0 & 0 & -\beta_{131}^C & 0 & 0 & 0 & 0 & 0 & 0 \\
0 & 0 & 0 & 0 & 0 & -\beta_{131}^C & 0 & 0 & 0 \\
\beta_{131}^C & 0 & 0 & 0 & \beta_{131}^C & 0 & 0 & 0 & 0
\end{array} \right) \quad (\text{R.9})$$

Q8

"Please double check this equation and other similar ones. The second e shall be in a complex conjugate form."

$$J_{i,PR}^{LCP} = \frac{I}{2} a_{PR} \beta_{ij}^C [\vec{e}_{LCP,h} \times \vec{e}_{LCP,h}] = \frac{I}{2\sqrt{3}} \begin{pmatrix} -\beta_{12}^C \\ -\beta_{12}^C \\ 0 \end{pmatrix}$$

A8

Action taken: We double-checked and updated the equations.

Q9

"In line 100, you claim that '...a 1/2 plate positioned after the 1/4 plate... the extracted LBPV contribution is shifted while the CBPV contribution remains unchanged'. Could you please clarify this result more? A 1/2 wave plate will switch the handedness of the light and you may expect a shifted CBPE?"

A9

A half-wave plate was ALWAYS positioned after the $\lambda/4$ -plate. We acknowledge that our attempt to simplify the experimental setup might have led to inaccurate description. We measured the CBPV response with and without half-wave plate:

Apparently, the change of handedness due to the half-wave plate is valid for all our experiments. We admit, that this aspect was unknown to us till indicated by Reviewer 2. As a result, earlier we had adjusted the CBPV tensor in Equation S10 ($\beta_{ij}^{C*} = -\beta_{ij}^C$):

$$\beta_{ij}^{C*} = \begin{pmatrix} 0 & -\beta_{12}^C & 0 \\ +\beta_{12}^C & 0 & 0 \\ 0 & 0 & 0 \end{pmatrix} \rightarrow \beta_{ij}^C = \begin{pmatrix} 0 & +\beta_{12}^C & 0 \\ -\beta_{12}^C & 0 & 0 \\ 0 & 0 & 0 \end{pmatrix} \quad (\text{R.10})$$

We have now removed this adjustment, which changes the sign of β_{12}^C in the entire the manuscript and supplementary info (Equation 2, 3, S10, S17-19, S21-23). Furthermore,

we had to simply interchange RCP and LCP in our experimental data (Figure 2c, 3b,
3c, 4, S4, S6, S9) and related texts.

**Actions taken:** We updated the manuscript and had to exchange LCP and RCP light.

Q10

*"To prove the existence of CD response in your samples, it would be more convinc-*
*ing if you simply conduct a test on a CD spectrometer. Ordinary CD spectrometer*
*can collect signals from UV to IR region, which perfectly covered the wavelength*
*used in your test and can provide valuable evidence."*

A10

We tried to point out the technical issues concerning absorption or CD spectroscopy in
our thin films in Reviewer 2 A5.

Q11

*"It would be helpful to estimate how much of PBV signal is canceled out by the*
*coexistence of two domain orientations, similar to the analysis in Ref. 4."*

A11

If we just have a look at the results from the tensorial analysis, there is an obvious
reduction of the current in \mathbf{x} -direction to zero e.g. comparing the overall CBPV response
(Eq. R.12) to a monodomain (Eq. R.11) response under RCP light:

CBPV response for RCP light ONLY L-configuration

$$j_{i,\text{P}_L}^{\text{RCP}} = I a_{\text{P}_L} \beta_{ij}^{\text{C}_i} [\vec{e}_{\text{RCP},h} \times \vec{e}_{\text{RCP},h}^*] = \frac{I}{\sqrt{3}} \begin{pmatrix} \beta_{12}^{\text{C}} \\ -\beta_{12}^{\text{C}} \\ 0 \end{pmatrix} \quad (\text{R.11})$$

Overall CBPV response for RCP light:

$$j_i^{\text{RCP}} = \frac{I}{2} \left[a_{\text{P}_R} (\beta_{ij}^{\text{C}_i} [\vec{e}_{\text{RCP},h} \times \vec{e}_{\text{RCP},h}^*]) + a_{\text{P}_L} (\beta_{ij}^{\text{C}_i} [\vec{e}_{\text{RCP},h} \times \vec{e}_{\text{RCP},h}^*]) \right] = \frac{I}{\sqrt{3}} \begin{pmatrix} 0 \\ -\beta_{12}^{\text{C}} \\ 0 \end{pmatrix} \quad (\text{R.12})$$

However, based on our results on domain-selective circular dichroism, there is even a
higher reduction of the resulting current, due to the fact of a differential absorption of
the light within the material.

**Q12**

*"What are the electrodes in Fig. 2a?"*

**A12**

The electrodes were structured using a standard photolithography process (rectangu-
lar electrode pairs $950 \times 400 \mu\text{m}^2$ with $\sim 40 \mu\text{m}$ gap). Platinum/Palladium alloy (Pt:Pd
80:20) was used as electrode material.

**Action taken:** We added the chapter "Device Fabrication" to the Methods section.

_____

**Q13**

*"Line 213, there may be a typo there: "The distance between the stoichiometric*
*The KrF excimer laser". The stoichiometric may be substrate?"*

**A13**

**Action taken:** We corrected the sentence to: "The distance between the stoichiometric
ceramic BiFeO_3 target and the substrate is set to 60 mm."

_____

**Q14**

*"Line 42, "analyzes" should be "analyze"."*

**A14**

**Action taken:** Corrected.

_____

**Q15**

*"There is a small typo in line 93 and line 98. The circularity of the light should*
*follow $\sin(2\theta)$ rather than $\sin(\theta)$."*

**A15**

**Action taken:** Corrected.

_____

Additional changes in the revised manuscript

- Schematic representation of electrodes $\pm 45^\circ$ had been mixed up in Fig. 4
- Supplementary Info: $J_i \rightarrow j_i$

References

- [1] S.-H. Baek, C. M. Folkman, J.-W. Park, S. Lee, C.-W. Bark, T. Tybell, and C.-B. Eom, “The nature of polarization fatigue in bifeo₃,” Advanced Materials, vol. 23, no. 14, pp. 1621–1625, 2011.
- [2] S. H. Baek, H. W. Jang, C. M. Folkman, Y. L. Li, B. Winchester, J. X. Zhang, Q. He, Y. H. Chu, C. T. Nelson, M. S. Rzchowski, X. Q. Pan, R. Ramesh, L. Q. Chen, and C. B. Eom, “Ferroelastic switching for nanoscale non-volatile magnetoelectric devices,” Nature materials, vol. 9, no. 4, pp. 309–314, 2010.
- [3] M.-M. Yang, Z.-D. Luo, D. J. Kim, and M. Alexe, “Bulk photovoltaic effect in monodomain bifeo₃ thin films,” Applied Physics Letters, vol. 110, no. 18, p. 183902, 2017.
- [4] D. Sando, A. Agbelele, D. Rahmedov, J. Liu, P. Rovillain, C. Toulouse, I. C. Infante, A. P. Pyatakov, S. Fusil, E. Jacquet, C. Carrétéro, C. Deranlot, S. Lisenkov, D. Wang, J.-M. Le Breton, M. Cazayous, A. Sacuto, J. Juraszek, A. K. Zvezdin, L. Bellaiche, B. Dkhil, A. Barthélémy, and M. Bibes, “Crafting the magnonic and spintronic response of bifeo₃ films by epitaxial strain,” Nature materials, vol. 12, no. 7, pp. 641–646, 2013.
- [5] A. Sander, M. Christl, C.-T. Chiang, M. Alexe, and W. Widdra, “Domain imaging on multiferroic bifeo₃ (001) by linear and circular dichroism in threshold photoemission,” Journal of Applied Physics, vol. 118, no. 22, p. 224102, 2015.
- [6] T. Choi, S. Lee, Y. J. Choi, V. Kiryukhin, and S.-W. Cheong, “Switchable ferroelectric diode and photovoltaic effect in bifeo₃,” Science (New York, N.Y.), vol. 324, no. 5923, pp. 63–66, 2009.
- [7] S. Y. Yang, L. W. Martin, S. J. Byrnes, T. E. Conry, S. R. Basu, D. Paran, L. Reichertz, J. Ihlefeld, C. Adamo, A. Melville, Y.-H. Chu, C.-H. Yang, J. L. Musfeldt, D. G. Schlom, J. W. Ager, and R. Ramesh, “Photovoltaic effects in bifeo₃,” Applied Physics Letters, vol. 95, no. 6, p. 062909, 2009.
- [8] S. Y. Yang, J. Seidel, S. J. Byrnes, P. Shafer, C.-H. Yang, M. D. Rossell, P. Yu, Y.-H. Chu, J. F. Scott, J. W. Ager, L. W. Martin, and R. Ramesh, “Above-bandgap voltages from ferroelectric photovoltaic devices,” Nature nanotechnology, vol. 5, no. 2, pp. 143–147, 2010.

- [9] A. Bhatnagar, A. Roy Chaudhuri, Y. Heon Kim, D. Hesse, and M. Alexe,
“Role of domain walls in the abnormal photovoltaic effect in bifeo₃,” Nature
communications, vol. 4, no. 1, p. 143, 2013.
- [10] J.-Y. Chauleau, T. Chirac, S. Fusil, V. Garcia, W. Akhtar, J. Tranchida,
P. Thibaudeau, I. Gross, C. Blouzon, A. Finco, M. Bibes, B. Dkhil, D. D. Khalyavin,
P. Manuel, V. Jacques, N. Jaouen, and M. Viret, “Electric and antiferromagnetic
chiral textures at multiferroic domain walls,” Nature materials, vol. 19, no. 4,
pp. 386–390, 2020.
- [11] M.-M. Yang, A. Bhatnagar, Z.-D. Luo, and M. Alexe, “Enhancement of local pho-
tovoltaic current at ferroelectric domain walls in bifeo₃,” Scientific reports, vol. 7,
p. 43070, 2017.
- [12] L. D. Barron, Molecular light scattering and optical activity. Cambridge: Cam-
bridge University Press, 2nd ed., rev. and enl ed., 2004.
- [13] I. Gross, W. Akhtar, V. Garcia, L. J. Martínez, S. Chouaieb, K. Garcia, C. Car-
rétéro, A. Barthélémy, P. Appel, P. Maletinsky, J.-V. Kim, J. Y. Chauleau,
481 N. Jaouen, M. Viret, M. Bibes, S. Fusil, and V. Jacques, “Real-space imaging
of non-collinear antiferromagnetic order with a single-spin magnetometer,” Nature,
483 vol. 549, no. 7671, pp. 252–256, 2017.
- [14] A. Haykal, J. Fischer, W. Akhtar, J.-Y. Chauleau, D. Sando, A. Finco, F. Godel,
Y. A. Birkhölzer, C. Carrétéro, N. Jaouen, M. Bibes, M. Viret, S. Fusil, V. Jacques,
and V. Garcia, “Antiferromagnetic textures in bifeo₃ controlled by strain and elec-
tric field,” Nature communications, vol. 11, no. 1, p. 1704, 2020.
- [15] S. M. Young, F. Zheng, and A. M. Rappe, “First-principles calculation of the bulk
photovoltaic effect in bismuth ferrite,” Physical review letters, vol. 109, no. 23,
p. 236601, 2012.
- [16] A. Talkenberger, I. Vrejoiu, F. Johann, C. Röder, G. Irmer, D. Rafaja, G. Schreiber,
492 J. Kortus, and C. Himcinschi, “Raman spectroscopic investigations of epitaxial bifeo
3 thin films on rare earth scandate substrates,” Journal of Raman Spectroscopy,
494 vol. 46, no. 12, pp. 1245–1254, 2015.
- [17] C. Himcinschi, J. Rix, C. Röder, M. Rudolph, M.-M. Yang, D. Rafaja, J. Kortus,
and M. Alexe, “Ferroelastic domain identification in bifeo₃ crystals using raman
spectroscopy,” Scientific reports, vol. 9, no. 1, p. 379, 2019.
- [18] D. W. Wilson, E. N. Glytsis, N. F. Hartman, and T. K. Gaylord, “Beam diame-
ter threshold for polarization conversion photoinduced by spatially oscillating bulk
photovoltaic currents in LiNbO₃:Fe,” Journal of the Optical Society of America B,
501 vol. 9, no. 9, p. 1714, 1992.

[19] J. F. Nye, Physical properties of crystals: Their representation by tensors and
matrices. Oxford science publications, Oxford: Clarendon Press, reprinted. ed.,
2012.

REVIEWER COMMENTS

Reviewer #1 (Remarks to the Author):

In my view the authors have carefully addressed all comments. I recommend publication.

Reviewer #2 (Remarks to the Author):

The authors have improved some parts of the manuscript significantly and I appreciate the authors' efforts. Again, I appreciate the interesting experimental results presented here and I enjoy reading them. However, I am still not convinced by the claims of circular dichroism. The major reason is that there is no any substantial experimental evidence to support this claim (see comments below). I understand the authors have difficulty to conduct the suggested experiments in my last comment. However, if the authors can revise the strong statements of CD (mainly to remove the strong statement of circular dichroism in abstract and conclusion and present CD as a possible explanation in the discussion part. The literature review can still be kept), I will be happy to see a revised manuscript further. My detailed technical comments are as follows.

1. CPGE tensor = Glass coefficient tensor multiplied by absorption coefficient ($\beta_{inm} = G_{inm}\alpha$). This means even you have a different photocurrent, you may have same absorption but different Glass coefficient tensor ($j_i = (\beta_{inm}^L e_n e_m^* + \beta_{in}^C \kappa_n)I$). There is no experimental observation of helicity-dependent absorption of light in domain variants in this manuscript (I appreciate the work on Raman scattering but that is not absorption).
2. For circular dichroism, the authors cite ref 5 (J. Appl. Phys. 118, 224102 (2015)) in the reply file to show that people have already confirmed the existence of circular dichroism experimentally. This is fine. However, clearly as stated in ref 5, the origin of circular dichroism is unknown (not magnetically related even, opposite to what the authors claim here). However, in refs. 10, 13 and 14, there exists chiral magnetic structure in BFO. The authors are suggested to discuss the argument here. Since this manuscript does not reveal the fundamental mechanism behind, I suggest the authors change the title to something like this "**The observation of abnormal circular photovoltaic effect in BiFeO₃ with 71° domain patterns**".

As follows include some statements that are not supported by the experiments. I have some comments here. There are also other sentences that have to be revised not quoted here.

3. The authors propose "opposite circular dichroism for the different polarization variants". This is too speculative.
4. This updated sentence "Our findings imply an intricate relation between the magnetic texture and the bulk photovoltaic effect in BiFeO₃ thin films" overstates the result since there is no any experimental demonstration of magnetic texture-BPE relation in this manuscript.
5. In these statements below "realize a condition wherein switch-like characteristic were observed, with light of opposite polarities resulting in '0' or ' $\pm V_{oc}$ '. As a result, we showcase a scenario wherein only half of the material contributes to the photovoltaic effect while the other half remains dormant. Analytical assessment with symmetry-specific tensors unravel a case of domain-selective absorption of CP light, in other words, circular dichroism.", following edits shall be made: "only half of the material contributes..." is not true since there is no experimental demonstration (this is a speculation); "domain-selective absorption of CP light" is not true since, again, there is no experimental demonstration (this is just a speculation).
6. This statement "The results evidently suggest an impact of the magnetic texture on the resultant photovoltaic response." is too speculative.

**Response to Reviewers:**
**Anomalous Circular Bulk Photovoltaic Effect in**
**BiFeO₃ Thin Films with Stripe-Domain Pattern**

David S. Knoche, Matthias Steimecke, Yeseul Yun,
Lutz Mühlenbein, Akash Bhatnagar

November 19, 2020

**1 Reviewer 1**

*"In my view the authors have carefully addressed all comments. I recommend*
*publication."*

We thank the Reviewer for positively assessing our work and for recommending our
work to be published.

**2 Reviewer 2**

*"The authors have improved some parts of the manuscript significantly and I*
*appreciate the authors' efforts. Again, I appreciate the interesting experimental*
*results presented here and I enjoy reading them. However, I am still not convinced*
*by the claims of circular dichroism. The major reason is that there is no any*
*substantial experimental evidence to support this claim (see comments below). I*
*understand the authors have difficulty to conduct the suggested experiments in*
*my last comment. However, if the authors can revise the strong statements of*
*CD (mainly to remove the strong statement of circular dichroism in abstract and*
*conclusion and present CD as a possible explanation in the discussion part. The*
*literature review can still be kept), I will be happy to see a revised manuscript*
*further. My detailed technical comments are as follows."*

***"The authors have improved some parts of the manuscript significantly and I***
***appreciate the authors' efforts. Again, I appreciate the interesting experimental***
***results presented here and I enjoy reading them. However, I am still not convinced***
***by the claims of circular dichroism. The major reason is that there is no any***
***substantial experimental evidence to support this claim (see comments below). I***
***understand the authors have difficulty to conduct the suggested experiments in***
***my last comment. However, if the authors can revise the strong statements of***
***CD (mainly to remove the strong statement of circular dichroism in abstract and***
***conclusion and present CD as a possible explanation in the discussion part. The***
***literature review can still be kept), I will be happy to see a revised manuscript***
***further. My detailed technical comments are as follows."***

We appreciate the positive feedback from the Reviewer and the suggestions for im-
provement. The reasons we persisted with our claims regarding circular dichroism were
based on the discrepancy between our tensorial analysis and the experimental data
which suggests domain-specific absorption of circularly polarized, in other words circu-
lar dichroism (CD). From our point of view, this assumption also seemed to reasonable as
CD has been already demonstrated in single crystal of BiFeO₃.⁽¹⁾ Importantly, the CD
was observed to be in direct relation with the domain arrangement. Our samples exhibit
even a superior domain arrangement with domain running across the entire thickness of
the films, and therefore a related CD is also possible. We totally agree that we do not
have any proof for CD, however, we performed Raman scattering experiments, which is
a proof for a differential interaction of the circularly polarized light.

The fundamental requirement for CD is chirality, which is also essential for Raman
optical activity (ROA). The spatially-resolved Raman scattering experiments, inspired
by ROA, clearly provided an image which is identical to the ferroelectric domain pattern.
While we agree that the Raman measurements do not consider absorption, but the
similarity in the results from Raman and PFM highlights the validity of chiral textures.
The subsequent attempts from our side to elaborate on the probable origin of the chiral
texture, and thus possibly CD, are mostly based on recent work based on samples of
analogues quality (in terms of epitaxial strain and domain arrangement). The related
explanations are not the primary objective of our work, which remains to be circular
bulk photovoltaic effect in BiFeO₃ and the associated anomalous behavior.

We agree that some of our statements regarding the CD in our manuscript are too
strong, especially since we do not have a direct proof of an absorption. In this regard,
we very much appreciate the Reviewer's understanding of the complexity of the desired
experiment. Henceforth, as per the suggestion of the Reviewer, we now have made
amends to the sentences wherein we might have made direct claims regarding CD.

_____,

Q1

***"CPGE tensor = Glass coefficient tensor multiplied by absorption coefficient ($\beta_{inn} =$***
***$G_{inn}\alpha$). This means even you have a different photocurrent, you may have same***

**absorption but different Glass coefficient tensor ($j_i = (\beta_{im}^L e_n e_m^* + \beta_{in}^C \kappa_n) I$). There is**
**no experimental observation of helicity-dependent absorption of light in domain**
**variants in this manuscript (I appreciate the work on Raman scattering but that is**
**not absorption)."**

**A1**

The Glass constant G were initially introduce to describe the observed photocurrents in
LiNbO₃ along the polar c-axis:(2, 3, 4)

$$j = \alpha G J, \quad (\text{R.1})$$

where α is the absorption coefficient and J the light intensity.

When it became clear that the BPV effect is of tensorial nature, a more general
equation was introduced:(4)

$$j_i = \beta_{ijl} e_j e_l^* J \quad (\text{R.2})$$

Under the assumption of isotropic absorption the scalar G can be replaced by G_{ijl} :(4)

$$G_{ijl} = \alpha^{-1} \beta_{ijl} \quad (\text{R.3})$$

In other words, under the assumption of isotropic absorption within the material,
the shape of tensor G_{ijl} and β_{ijl} is equivalent (since α is just a scalar in this case). The
calculated response, with glass coefficient tensor or BPV tensor, results in the same pho-
tovoltaic current (again, under the assumption of isotropic absorption). Therefore, the
discrepancy between the calculated response and the experimental data can be addressed
with differential (anisotropic) absorption.

We agree that our work describes the observation of a anomalous circular bulk pho-
voltaic response. We do not have experimental proof of domain-dependent absorption of
CP light (CD behavior). Our assumption of CD, as stated above, is based on the quali-
tative tensor analysis of each domain variants, strengthened by using different electrodes
geometries and Raman scattering experiments.

**Q2**

**"For circular dichroism, the authors cite ref 5 (J. Appl. Phys. 118, 224102 (2015))**
**in the reply file to show that people have already confirmed the existence of circular**
**dichroism experimentally. This is fine. However, clearly as stated in ref 5, the origin**
**of circular dichroism is unknown (not magnetically related even, opposite to what**
**the authors claim here). However, in refs. 10, 13 and 14, there exists chiral**
**magnetic structure in BFO. The authors are suggested to discuss the argument**
**here. Since this manuscript does not reveal the fundamental mechanism behind,**
**I suggest the authors change the title to something like this "The observation of**
**abnormal circular photovoltaic effect in BiFeO3 with 71° domain patterns".**

***As follows include some statements that are not supported by the experiments. I***
***have some comments here. There are also other sentences that have to be revised***
***not quoted here."***

**A2**

There are two aspects of this discussion. First, CD in BiFeO₃ which has been clearly
demonstrated with PEEM, as acknowledged by the Reviewer as well. Importantly, the
CD was found to be in direct relation with the domain pattern. So, an analogous situa-
tion can be also realistically assumed in our samples which have even a superior domain
arrangement. And second, the origin of CD which we agree in (1) (***ref 5***) was not
attributed to the magnetic structure. The Néel temperature of the sample (independ-
ently measured in a SQUID setup) was found to be around ~620 K which typically
has been reported to be around ~640 K(5). In addition, the PEEM investigations were
conducted at ~670 K. The involved temperatures are in close vicinity of one another,
and position of thermo-couples at different locations near the samples, and in differ-
ent setups, is sufficient to cause these slight changes in measured temperature. Also,
a temperature gradient across the sample thickness cannot be ruled out, specifically if
the measurements are surface sensitive. We must explicitly emphasize here that we are
in absolutely no position to comment on the technical credibility of the their work, and
nor it is our intention. Our objective here is simply to state few scenarios which could
influence an accurate temperature measurement.

The existence of a chiral magnetic structure in BiFeO₃ has been demonstrated using
neutron diffraction (6, 7, 8, 9, 10), Mössbauer spectroscopy (11), low-energy Raman
scattering experiments (11, 12), nuclear resonant scattering (12), scanning nitrogen-
vacancy magnetometry(13, 14, 15), resonant elastic X-ray scattering (14, 15), and X-ray
diffraction measurements (10). The cycloid has been demonstrated in both single crystals
and thin films. A nice overview is provided in a review recently written by Burns et al.
(16).

Gross et al.(13) (***ref 13***) were the first to visualize antiferromagnetic spin cycloid
in BiFeO₃ thin films with 71° domain pattern using scanning nitrogen vacancy magne-
tometry in real space. The same technique was used by Chauleau et al.(14) (***ref 10***)
and Haykal et al. (15) (***ref 14***). Additionally, both works concluded the existence of
the spin cycloid from an observed circular dichroic behavior in their resonant elastic
X-ray scattering (REXS) measurements. Therefore, the spin cycloid as origin of circular
dichroic behavior has been demonstrated before.

**Action taken:** Overall we agree with the Reviewer for the change of the title. As per
the suggestion, the new title and abstract are more reflective of our main findings.

**Q3**

*"The authors propose "opposite circular dichroism for the different polarization*
*variants". This is too speculative."*

**A3**

**Action taken:** We adjusted the corresponding statements in the revised manuscript.

**Q4**

*"This updated sentence "Our findings imply an intricate relation between the*
*magnetic texture and the bulk photovoltaic effect in BiFeO3 thin films" overstates*
*the result since there is no any experimental demonstration of magnetic texture-*
*BPE relation in this manuscript."*

**A4**

**Action taken:** We removed this sentence in the revised manuscript.

**Q5**

*"In these statements below "realize a condition wherein switch-like characteristic*
*were observed, with light of opposite polarities resulting in '0' or ' \pm Voc'. As a*
*result, we showcase a scenario wherein only half of the material contributes to the*
*photovoltaic effect while the other half remains dormant. Analytical assessment*
*with symmetry-specific tensors unravel a case of domain-selective absorption of*
*CP light, in other words, circular dichroism.", following edits shall be made: "only*
*half of the material contributes... " is not true since there is no experimental*
*demonstration (this is a speculation); "domain-selective absorption of CP light"*
*is not true since, again, there is no experimental demonstration (this is just a*
*speculation)."*

**A5**

**Action taken:** We restructured our Abstract and reduced the content to fulfill the
required word limitation.

**Q6**

*"This statement "The results evidently suggest an impact of the magnetic texture*
*on the resultant photovoltaic response." is too speculative."*

A6

Action taken: We removed this sentence in the revised manuscript.

3 Changes in the revised manuscript

The changes in the revised manuscript are blue-colored.

References

- [1] A. Sander, M. Christl, C.-T. Chiang, M. Alexe, and W. Widdra, “Domain imaging on multiferroic bifeo 3 (001) by linear and circular dichroism in threshold photoemission,” Journal of Applied Physics, vol. 118, no. 22, p. 224102, 2015.
- [2] A. M. Glass, D. von der Linde, and T. J. Negran, “High-voltage bulk photovoltaic effect and the photorefractive process in linbo 3,” Applied Physics Letters, vol. 25, no. 4, pp. 233–235, 1974.
- [3] A. M. Glass, D. von der Linde, D. H. Auston, and T. J. Negran, “Excited state polarization, bulk photovoltaic effect and the photorefractive effect in electrically polarized media,” Journal of Electronic Materials, vol. 4, no. 5, pp. 915–943, 1975.
- [4] P. J. Sturman and V. M. Fridkin, Photovoltaic and Photo-refractive Effects in Noncentrosymmetric Materials. Taylor & Francis Ltd, 1992.
- [5] S. Kiselev, R. P. Ozerov, and G. S. Zhdanov, “Detection of magnetic order in ferroelectric bifeo 3 by neutron diffraction,” Soviet physics. Doklady, vol. 7, p. 742, 1963.
- [6] I. Sosnowska, T. P. Neumaier, and E. Steichele, “Spiral magnetic ordering in bismuth ferrite,” Physics Letters A, vol. 15, no. 23, pp. 4835–4846, 1982.
- [7] H. Béa, M. Bibes, S. Petit, J. Kreisel, and A. Barthélémy, “Structural distortion and magnetism of bifeo 3 epitaxial thin films: A raman spectroscopy and neutron diffraction study,” Philosophical Magazine Letters, vol. 87, no. 3-4, pp. 165–174, 2007.
- [8] D. Lebeugle, D. Colson, A. Forget, M. Viret, A. M. Bataille, and A. Gukasov, “Electric-field-induced spin flop in bifeo3 single crystals at room temperature,” Physical review letters, vol. 100, no. 22, p. 227602, 2008.
- [9] M. Ramazanoglu, M. Laver, W. Ratcliff, S. M. Watson, W. C. Chen, A. Jackson, K. Kothapalli, S. Lee, S.-W. Cheong, and V. Kiryukhin, “Local weak ferromagnetism in single-crystalline ferroelectric bifeo3,” Physical review letters, vol. 107, no. 20, p. 207206, 2011.

- [10] J. Bertinshaw, R. Maran, S. J. Callori, V. Ramesh, J. Cheung, S. A. Danilkin, W. T.
Lee, S. Hu, J. Seidel, N. Valanoor, and C. Ulrich, “Direct evidence for the spin
cycloid in strained nanoscale bismuth ferrite thin films,” Nature communications,
205 vol. 7, p. 12664, 2016.
- [11] D. Sando, A. Agbelele, D. Rahmedov, J. Liu, P. Rovillain, C. Toulouse, I. C. In-
fante, A. P. Pyatakov, S. Fusil, E. Jacquet, C. Carrétéro, C. Deranlot, S. Lisenkov,
D. Wang, J.-M. Le Breton, M. Cazayous, A. Sacuto, J. Juraszek, A. K. Zvezdin,
209 L. Bellaiche, B. Dkhil, A. Barthélémy, and M. Bibes, “Crafting the magnonic and
210 spintronic response of bifeo3 films by epitaxial strain,” Nature materials, vol. 12,
no. 7, pp. 641–646, 2013.
- [12] A. Agbelele, D. Sando, C. Toulouse, C. Paillard, R. D. Johnson, R. Ruffer, A. F.
Popkov, C. Carrétéro, P. Rovillain, J.-M. Le Breton, B. Dkhil, M. Cazayous,
Y. Gallais, M.-A. Méasson, A. Sacuto, P. Manuel, A. K. Zvezdin, A. Barthélémy,
215 J. Juraszek, and M. Bibes, “Strain and magnetic field induced spin-structure tran-
216 sitions in multiferroic bifeo3,” Advanced materials (Deerfield Beach, Fla.), vol. 29,
no. 9, 2017.
- [13] I. Gross, W. Akhtar, V. Garcia, L. J. Martínez, S. Chouaieb, K. Garcia, C. Car-
rétéro, A. Barthélémy, P. Appel, P. Maletinsky, J.-V. Kim, J. Y. Chauleau,
220 N. Jaouen, M. Viret, M. Bibes, S. Fusil, and V. Jacques, “Real-space imaging
of non-collinear antiferromagnetic order with a single-spin magnetometer,” Nature,
222 vol. 549, no. 7671, pp. 252–256, 2017.
- [14] J.-Y. Chauleau, T. Chirac, S. Fusil, V. Garcia, W. Akhtar, J. Tranchida,
P. Thibaudeau, I. Gross, C. Blouzon, A. Finco, M. Bibes, B. Dkhil, D. D. Khalyavin,
P. Manuel, V. Jacques, N. Jaouen, and M. Viret, “Electric and antiferromagnetic
chiral textures at multiferroic domain walls,” Nature materials, vol. 19, no. 4,
pp. 386–390, 2020.
- [15] A. Haykal, J. Fischer, W. Akhtar, J.-Y. Chauleau, D. Sando, A. Finco, F. Godel,
Y. A. Birkhölzer, C. Carrétéro, N. Jaouen, M. Bibes, M. Viret, S. Fusil, V. Jacques,
and V. Garcia, “Antiferromagnetic textures in bifeo3 controlled by strain and elec-
tric field,” Nature communications, vol. 11, no. 1, p. 1704, 2020.
- [16] S. R. Burns, O. Paull, J. Juraszek, V. Nagarajan, and D. Sando, “The experimen-
talist’s guide to the cycloid, or noncollinear antiferromagnetism in epitaxial bifeo3,”
Advanced Materials, p. e2003711, 2020.

REVIEWERS' COMMENTS

Reviewer #2 (Remarks to the Author):

I am happy with the updated manuscript and statements. I suggest publishing this manuscript now.